# CALM: Consensus-Aware Localized Merging for Multi-Task Learning

Kunda Yan [* 1]   Min Zhang [* 2]   Sen Cui [† 1]   Zikun Qu [3]   Bo Jiang [2]   Feng Liu [4]   Changshui Zhang [† 1]

## Abstract

Model merging aims to integrate the strengths of multiple fine-tuned models into a unified model while preserving task-specific capabilities. Existing methods, represented by task arithmetic, are typically classified into global- and local-aware methods. However, global-aware methods inevitably cause parameter interference, while local-aware methods struggle to maintain the effectiveness of task-specific details in the merged model. To address these limitations, we propose a **C**onsensus-**A**ware **L**ocalized **M**erging (CALM) method which incorporates **localized information aligned with global task consensus**, ensuring its effectiveness post-merging. CALM consists of three key components: (1) **class-balanced entropy minimization sampling**, providing a more flexible and reliable way to leverage unsupervised data; (2) **an efficient-aware framework**, selecting a small set of tasks for sequential merging with high scalability; (3) **a consensus-aware mask optimization**, aligning localized binary masks with global task consensus and merging them conflict-free. Experiments demonstrate the superiority and robustness of our CALM, significantly outperforming existing methods and achieving performance close to traditional MTL.

## 1. Introduction

Multi-task learning (MTL) facilitates knowledge transfer across different tasks through a shared backbone, improving both model efficiency and performance (Liu et al., 2019a;b; Dong et al., 2015; Yang et al., 2024c; Qiu et al., 2024; Xin et al., 2024). This approach has been widely applied in fields such as computer vision (Cai et al., 2024a; Lu et al., 2020; Qiu et al., 2024; Yang et al., 2024c; Cai et al., 2024b), natural language processing (Liu et al., 2019b; Dong et al., 2015; Audibert et al., 2023; Xin et al., 2024), and so on. However, in the context of large foundational models, traditional MTL methods often require the centralized collection and processing of vast amounts of data, leading to high costs in data labeling and computational resources. At the same time, with the widespread use of pre-trained models, downstream tasks typically fine-tune the same pre-trained model (*e.g.*, ViT (Dosovitskiy, 2020) or BERT (Devlin, 2018)) independently, and the fine-tuned models are usually released without disclosing the specifics of the original training data to protect privacy. In recent years, researchers have changed their focus towards *exploring how to effectively integrate multiple independently trained models to achieve MTL without re-training on the original data*.

In response, model merging (or fusion) methods have been employed to address challenges in traditional MTL (Yang et al., 2024b; Yadav et al., 2023; Ilharco et al., 2023; Tang et al., 2023; Yang et al., 2024a; Wang et al., 2024; He et al., 2024; Chen & Kwok, 2024). A key approach in this field is **task arithmetic** (Ilharco et al., 2023), which introduces the **task vector** to represent task-specific weight adjustments. A task vector is created by subtracting task-specific weights from pre-trained weights, producing a unique representation for each task. Research has demonstrated that by strategically merging multiple task vectors and incorporating them into a pre-trained model, a new model can be created to effectively support MTL. As a result, most model merging methods rely on task vectors to construct integrated models.

Task vector-based model merging methods can currently be categorized into two groups based on the information focus between task vectors: global-aware methods (as illustrated in Figure 1(a)) and localized-aware methods (as illustrated in Figure 1(b)). Global-aware methods, such as those utilizing arithmetic mean (Ilharco et al., 2023) and learned merging weights (Yang et al., 2024b), process the models in **a global manner**, performing arithmetic operations on **all the parameters of the fine-tuned models**. However, parameters from different fine-tuned models inevitably inter-

*Equal contribution † Corresponding author [1]Institute for Artificial Intelligence, Tsinghua University (THUAI), Beijing National Research Center for Information Science and Technology (BNRist), Department of Automation, Tsinghua University, Beijing, P.R.China [2]East China Normal University (ECNU) [3]The Chinese University of Hong Kong, Shenzhen [4]School of Computing and Information Systems, The University of Melbourne. Correspondence to: Sen Cui <cuis@mail.tsinghua.edu.cn>, Changshui Zhang <zcs@mail.tsinghua.edu.cn>.

*Proceedings of the 42nd International Conference on Machine Learning*, Vancouver, Canada. PMLR 267, 2025. Copyright 2025 by the author(s).

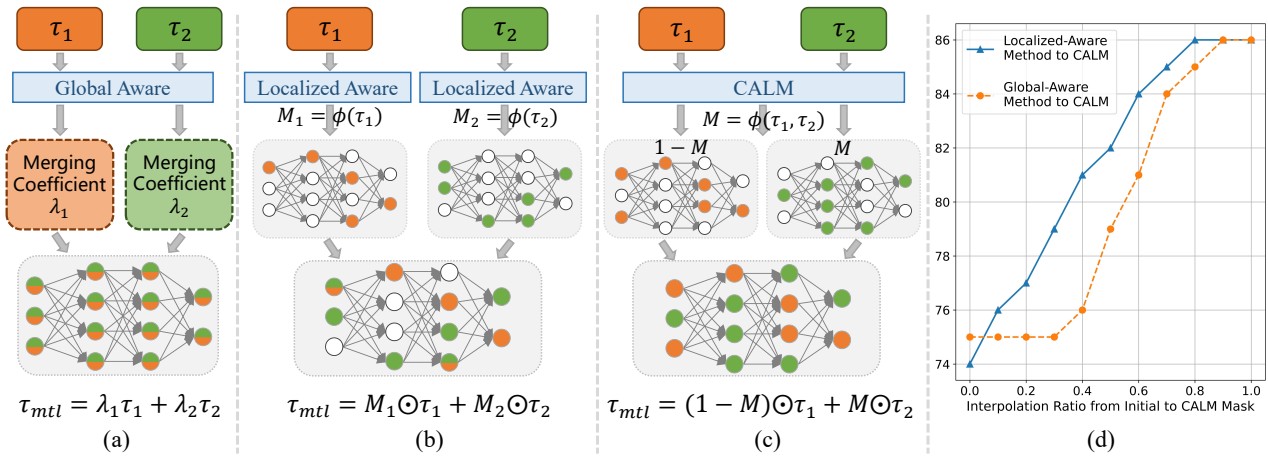

*Figure 1.* (a) Global-aware methods: Process the models holistically by applying arithmetic operations to all parameters of the fine-tuned models. (b) Localized-aware methods: Rely on task-specific optimized masks to extract local information from task vectors. (c) Our CALM method: Utilize masks to extract locally effective information with global task consensus and merge it without information conflict. (d) Experimental validation: Present the accuracy curve of the merged model which gradually interpolates the information extracted by global-aware and localized-aware methods with our approach, validating that CALM integrates more effective localized information.

fere with each other, which leads to suboptimal performance in the merged model. Prateek et al (Yadav et al., 2023) validates that redundant parameter updates during fine-tuning often serve as a significant source of conflicts.

In contrast, localized-aware methods extract local information from task vectors, which mitigates conflicts between global parameters. The core of these methods lies in **identifying effective local information**, with criteria varying across approaches. For instance, Ties-merging (Yadav et al., 2023) emphasizes parameters with larger magnitudes, while Localize-and-stitch (He et al., 2024) prioritizes parameters that preserve local task performance. Although these methods ensure strong local task performance before merging, the overall performance of the merged model tends to degrade significantly, indicating that this local information may not be universally effective for all tasks. **This naturally raises a pivotal question**:

> *How to identify and extract effective information in model merging to enhance performance on all tasks?*

Given the above analysis, we propose that effective information in model merging should possess two key characteristics: (1) effective information can be **represented by localized parameters**, minimizing interference from global merging; and (2) effective information should **align with the global task consensus**, maintaining its efficacy in the merged model. Based on this criterion, we propose a **C**onsensus-**A**ware **L**ocalized **M**erging (CALM) method. The core idea of CALM is illustrated in Figure 1 (c). In merging two task vectors, a mask $M$ extracts effective information from each task vector, which is optimized through the global tasks, ensuring alignment with the global consen-

sus. During merging, the effective information from both task vectors is merged without conflict. Figure 1 (d) presents experimental validation. By interpolating the information extracted by global-aware and localized-aware methods with our approach, we observe that as the merged model's information approaches ours, its performance improves. This confirms the superiority of the global consensus-based information extraction in model merging.

Specifically, CALM consists of three key components: First, we introduce a **class-balanced entropy minimization sampling** method. Unlike minimizing the entropy of unsupervised data or using supervised training data, this method provides a more flexible and reliable way to leverage unsupervised data. Second, we propose an **efficient-aware framework**, which only requires selecting a small number of tasks for sequential merging, offering a more efficient solution with good scalability. Finally, we present a **consensus-aware mask optimization** method, which locates masks with global consensus using credible datasets and merges them without conflict. Experimental results demonstrate the superiority and robustness of our approach, significantly outperforming existing baseline methods and approaching the performance of traditional MTL. Code is available at https://github.com/yankd22/CALM.

Our main contributions of this paper are three-fold:

- We explore a core issue in model merging: how to identify and extract effective information on each task. We argue that effective information should be localized and aligned with global task consensus.

- We propose a novel model merging framework, CALM, which leverages reliable unsupervised data to extract

local information aligned with global task consensus, and enables efficient and conflict-free merging.

- We conduct extensive experiments on various datasets and demonstrate that CALM exhibits superior performance in model merging, surpassing other state-of-the-art global-aware and localized-aware methods.

## 2. Related Work

### 2.1. Model Merging for Multi-Task Learning

Model merging represents the process of combining multiple independent models into a single model (Ilharco et al., 2023), with two primary application scenarios: (1) merging models trained on the same task to enhance the final model's resistance to forgetting and improve its generalization (Wortsman et al., 2022; Yao et al., 2023; Zhu et al., 2024; Panigrahi et al., 2023). (2) merging models trained on different tasks to achieve MTL (Yang et al., 2024b; He et al., 2024; Matena & Raffel, 2022; Yu et al., 2024; Yang et al., 2024a), which is more aligned with real-world applications and is the focus of this paper. These MTL model merging methods can currently be categorized into two groups based on the information focus between task vectors: global-aware methods and localized-aware methods.

**Global-aware methods.** Global-aware methods, such as those employing arithmetic mean (Ilharco et al., 2023) or learned merging weights (Yang et al., 2024b), process models holistically by applying arithmetic operations across all parameters of fine-tuned models. For instance, AdaMerging (Yang et al., 2024b) leverages unlabeled test data to automatically learn merging coefficients at the task or layer level. Similarly, DARE (Yu et al., 2024) reduces redundant neuron updates before scaling neurons for merging. However, combining parameters from different fine-tuned models inevitably leads to interference with each other, resulting in suboptimal performance in the merged model.

**Localized-aware methods.** In contrast, localized-aware methods (Yadav et al., 2023; He et al., 2024; Wang et al., 2024; Chen & Kwok, 2024) emphasize extracting task-specific, localized information from task vectors to reduce conflicts among global parameters. These approaches center on selecting effective local information, with selection criteria differing between methods. For example, (Yadav et al., 2023) prioritizes parameters with larger magnitudes, whereas (He et al., 2024) focuses on parameters that maintain local task performance. While these methods help preserve strong local task performance before merging, the overall performance of the merged model often suffers, suggesting that this localized information may not generalize effectively across tasks. Therefore, we propose that the effective information extracted in model merging should be localized information with global task consensus.

### 2.2. Traditional Multi-Task Learning

Traditional multi-task learning (MTL) improves model performance and efficiency by transferring knowledge across multiple tasks through a shared backbone (Liu et al., 2019b; Dong et al., 2015; Yang et al., 2024c; Zhang et al., 2024; 2023; 2022). However, MTL often faces the problem of negative transfer due to conflicts and interference between tasks (Lin et al., 2019; 2020). Current research addressing negative transfer primarily focuses on two areas: (1) The classic SharedBottom architecture (Caruana, 1997) performs poorly when task correlations are weak, whereas modern architectures alleviate negative transfer through modular design (Ponti et al., 2022; Chen et al., 2023; Swamy et al., 2024), sparsification (Sun et al., 2022; Calandriello et al., 2014; Ma et al., 2022), and soft sharing of the backbone network (Shi et al., 2023). (2) Other research tackles task interference from an optimization perspective, such as adjusting the loss weights for each task (Yu et al., 2020; Liu et al., 2021), resolving conflicts in multi-task gradient directions or signs, or suppressing the dominance of learning rates and gradients (Chen et al., 2018). Unlike these traditional methods that focus on loss weights or gradient space, our proposed CALM improves MTL performance and addresses conflicts through emphasizing local information aligned with global consensus in model merging.

## 3. Preliminaries

In this section, we first give the notation and definition, then provide a detailed explanation of model merging solutions.

**Notation.** Let the neural network model be denoted as $f : \mathcal{X} \times \Theta \mapsto \mathcal{Y}$, where the parameters are represented by $\theta \in \Theta \in \mathbb{R}^n$, the input by $x_i \in \mathcal{X} \in \mathbb{R}^d$, and output by $y_i \in \mathcal{Y} \in \mathbb{R}^c$. Here, $n$ denotes the number of parameters, $d$ is the input dimension, and $c$ means the number of output classes. Consider a scenario involving $T$ independent tasks, where each task $t$ has an associated training dataset $\mathcal{D}_{tr}^t(\mathcal{X}, \mathcal{Y})$. Given a pre-trained model $\theta_{pre}$, such as ViT (Dosovitskiy, 2020) or BERT (Devlin, 2018), each task's training dataset is used to fine-tune the pre-trained model, resulting in $T$ fine-tuned models, denoted as $\{\theta_{ft}^t\}_{t=1}^T$.

**Traditional multi-task learning.** Model merging is to merge the weights $\{\theta_{ft}^t\}_{t=1}^T$ into a unified model $\theta_{mtl}$ without retraining, such that the combined model performs effectively across all tasks. Specifically, given the test datasets $\mathcal{D}_{te}^t$ for each task $t$, the goal is to minimize the empirical loss $\mathcal{L}$ of the merged model $\theta_{mtl}$, defined as:

$$\mathcal{L}(\theta_{mtl}) = \frac{1}{T} \sum_{t=1}^{T} \frac{1}{|\mathcal{D}_{te}^t|} \sum_{(x_i, y_i) \in \mathcal{D}_{te}^t} l(f_{\theta_{mtl}}(x_i), y_i), \quad (1)$$

where $l(\cdot)$ is the loss function, e.g., cross-entropy.

**Task vector-based model merging.** Following task arithmetic (Ilharco et al., 2023), the task vector $\tau_t$ is defined as the difference between the fine-tuned model parameters $\theta_{ft}^t$ and the pretrained model parameters $\theta_{pre}$, i.e., $\tau_t = \theta_{ft}^t - \theta_{pre}$. It captures the model parameter updates during fine-tuning, reflecting adjustments for task adaptation. Task vector plays a crucial role in model merging, providing a compact yet highly informative representation of task-specific modifications that facilitates efficient MTL.

**Representative model merging solutions.**

- **Task Arithmetic** (Ilharco et al., 2023) is a global-aware approach that merges task vectors with the pretrained model $\theta_{pre}$ using weighted coefficients $\lambda$. This allows for a significant improvement over simple averaging. The formula is $\theta_{mtl} = \theta_{pre} + \lambda \sum_{t=1}^{T} \tau_t$.

- **Ties-Merging** (Yadav et al., 2023) is a localized-aware method that focuses on the parameters with the largest magnitudes in the task vectors. It employs the elect strategy to selectively merge these parameters with minimal conflict, resulting in a merged parameter $\delta_{ties}$. The formula is $\theta_{mtl} = \theta_{pre} + \lambda \delta_{ties}$.

- **Adamerging** (Yang et al., 2024b) is a global-aware method that optimizes the merging weights of task vectors by minimizing the entropy of unsupervised samples. This approach refines the merging process across tasks and allows for layer-level adjustments. The formula is $\theta_{mtl} = \theta_{pre} + \sum_{t=1}^{T} \lambda_t \tau_t$.

## 4. Methodology

This section details the implementation of CALM. Section 4.1 explains the method for obtaining a reliable unsupervised dataset, while Section 4.2 introduces the efficient-aware framework, which extracts only a small number of task execution sequences for merging. Building on this, Section 4.3 presents a method for identifying localized parameters with global task consensus using binary masks. The overall process is summarized in Algorithm 1.

### 4.1. Class-Balanced Entropy Minimization Sampling

Utilizing additional sample information to aid model merging often significantly improves performance. For example, Adamerging (Yang et al., 2024b) learns from the entropy of unsupervised samples, allowing for balanced task performance, while Localize-and-Stitch (He et al., 2024) relies on training set data to identify effective parameter points. In contrast, acquiring a subset of unsupervised samples is more practical in real-world scenarios.

However, we argue that the entropy values used in Adamerging (Yang et al., 2024b) are not entirely reliable, as shown in Figure 8, and may be affected by the merged model. Inac-

curate entropy estimates can misdirect optimization, leading to suboptimal merged models. To address this, we propose class-balanced entropy minimization sampling, which adaptively selects credible samples, using pseudo-labels to ensure reliable information remains intact during merging.

**Shannon entropy measure.** Given a dataset $\mathcal{D} = \{x_i\}_{i=1}^N$ and a finetuned model with parameters $\theta$, the model's logit output $P(\hat{y}|x_i, \theta)$ provides the class probabilities. The Shannon entropy (Shannon, 1948) for each sample is defined as:

$$H(x_i, \theta) = -\sum_{c=1}^{C} P(\hat{y} = c|x_i, \theta) \log P(\hat{y} = c|x_i, \theta), \quad (2)$$

where $C$ is the total number of classes. High entropy indicates greater uncertainty in the model's prediction, while low entropy suggests higher confidence.

**Entropy minimization sampling (EMS).** EMS aims to select samples with the lowest entropy to construct a credible sample set from the unsupervised dataset. This approach is widely used in fields like active learning (Wu et al., 2022; Xie et al., 2022) and anomaly detection (Yoon et al., 2023) to reduce uncertainty and enhancing model stability and robustness. Here, we apply EMS to model merging.

Given multiple task-specific unsupervised datasets $\{\mathcal{D}^t(\mathcal{X})\}_{t=1}^T$, our goal is to select low-entropy samples for each task $t$, which can be formulated as:

$$\hat{\mathcal{D}}^t = \arg\min_{\mathcal{D}^t} \sum_{x_i \in \mathcal{D}^t} H(x_i, \theta_{\text{ft}}^t). \quad (3)$$

This selection improves model merging robustness at the data level. Additionally, to prevent interference with entropy information during parameter merging, we assign pseudo-labels to unsupervised data based on logits:

$$\hat{y}_i = \arg\max_c P(\hat{y} = c|x_i, \theta). \quad (4)$$

Thus, we obtain a credible dataset with pseudo-labels that remain unaffected when merging, denoted as $\{\hat{\mathcal{D}}^t(\mathcal{X}, \hat{\mathcal{Y}})\}_{t=1}^T$.

**Class-balanced entropy minimization sampling (CB-EMS).** To address class imbalance that may occur from selecting only low-entropy samples, we propose class-balanced entropy minimization sampling (CB-EMS). The core idea is to select an equal number of low-entropy samples from each class to ensure balanced representation. For the dataset of class $c$, $\mathcal{D}_c^t$, we define:

$$\hat{\mathcal{D}}_{cb}^t = \bigcup_{c=1}^{C} \{x_i \in \mathcal{D}_c^t | H(x_i, \theta_{ft}^t) \leq H_{(k)}(\mathcal{D}_c^t, \theta_{ft}^t)\}, \quad (5)$$

where $H_{(k)}(\mathcal{D}_c^t, \theta_{ft}^t)\}$ is the entropy of the $k$-th lowest entropy sample in $\mathcal{D}_c^t$. Finally, we assign pseudo-labels to create the class-balanced credible dataset $\{\hat{\mathcal{D}}_{cb}^t(\mathcal{X}, \hat{\mathcal{Y}})\}_{t=1}^T$.

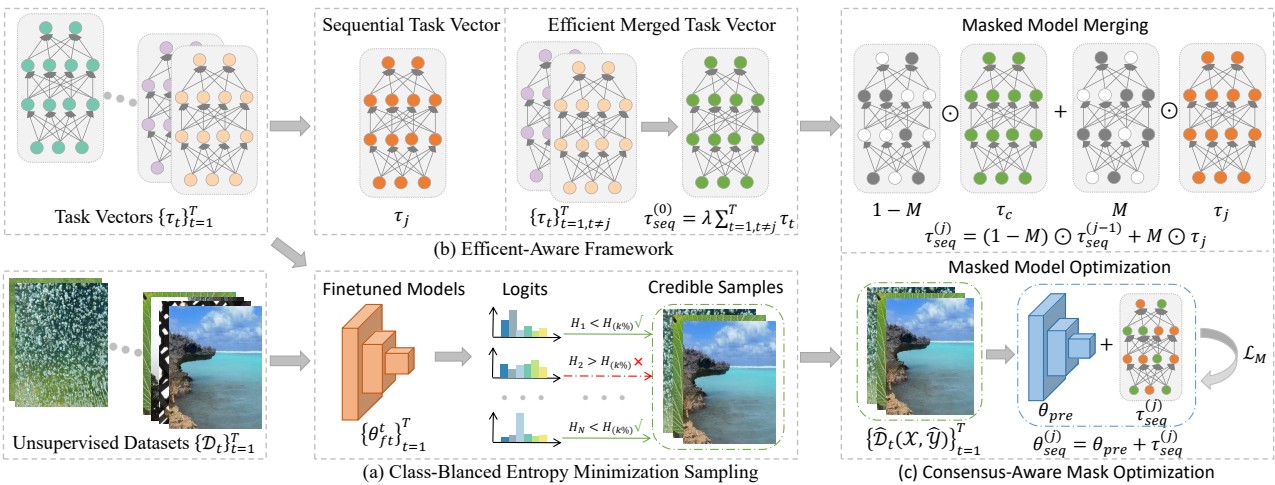

*Figure 2.* Illustration of Our CALM Approach. CALM involves three stages: (a) Efficient Aware: Task vectors are split, with most merged efficiently into $\tau_c$ via Task Arithmetic, and a smaller part used for sequential merging; (b) Class-Balanced Entropy Minimization Sampling: Finetuned models extract credible samples from the unlabeled dataset while maintaining class balance; (c) Consensus-Aware Mask Optimization: Sequential tasks are merged into the $\tau_c$ model using a mask, which is then refined using the credible samples.

### 4.2. Efficient-Aware Framework

We propose an efficient-aware model merging framework. Unlike traditional methods, for a given set of tasks, represented by task vectors $\{\tau_t\}_{t=1}^T$, we divide these vectors into two parts: $S$ and $\overline{S}$, with the majority of tasks belonging to $S$. For the tasks in $S$, we perform efficient merging utilizing task arithmetic (Ilharco et al., 2023), resulting in an initial merged model defined as $\tau_c = \lambda \sum_{i \in S} \tau_i$.

The remaining tasks in $\overline{S}$ are merged sequentially. We use $\tau_c$ as the base model, denoted as $\tau_{\text{seq}}^{(0)}$, and then incrementally merge each task from $\overline{S}$. For each task $j \in \overline{S}$, the current merged model $\tau_{\text{seq}}^{(j-1)}$ is updated by merging with the task vector $\tau_j$, resulting in $\tau_{\text{seq}}^{(j)} = f(\tau_{\text{seq}}^{(j-1)}, \tau_j)$, where $f(\cdot)$ is an applicable model merging method. This process is repeated until all sequential tasks have been merged, yielding the final merged task vector $\tau_{\text{mtl}}$. Figure 2(b) illustrates the merging process when $\overline{S}$ contains only one model.

The efficient-aware framework demonstrates substantial advantages, leading to a significant enhancement in merging efficiency, while simultaneously mitigating the rapid increase in computational complexity as the number of tasks expands. Therefore, it shows impressive scalability, particularly when dealing with an increasing volume of tasks.

Furthermore, our framework shows high flexibility in handling new tasks. In practical scenarios, task asynchrony is very common, with different tasks often completing at different times. Traditional methods typically require all tasks to be completed before merging, which inevitably leads to delays. Additionally, in resource-constrained environments, it is challenging to merge all tasks simultaneously using conventional approaches. In contrast, our framework effi-

ciently integrates new tasks without needing to recombine all previous tasks, thereby showing greater practicality.

### 4.3. Consensus-Aware Mask Optimization

Based on the CB-EMS dataset and the efficient-aware framework, this part aims to address how to identify and extract effective information during the model merging process. As demonstrated by the validation experiments in Figure 1, effective information is not uniformly present across all parameter points. Instead, extracting appropriate localized parameters can significantly improve the performance of model merging. Furthermore, the extraction of effective information should not be limited to individual tasks; due to the interference among tasks in model merging, the effectiveness should be the consensus of all tasks globally.

Building on this understanding, we propose two guiding principles for our design approach: First, model merging should focus on localized parameters. Second, the selection of these localized parameters should take into account global task information to achieve consensus.

**Mask is an effective tool for focusing on local parameters.** Several existing studies have explored the idea of using masks. For instance, Ties-Merging (Yadav et al., 2023) approximates the selection of the largest parameter points through a mask, while Localize-and-Stitch (He et al., 2024) aims to find a mask that better represents a single task. In this study, we propose a novel masked model merging strategy. Given a current merged task vector $\tau_{seq}^{(j-1)}$, a sequential task vector $\tau_j$ awaiting merging, and a binary mask $M$, we define the masked model merging as follows:

$$\tau_{seq}^{(j)} = (1 - M) \odot \tau_{seq}^{(j-1)} + M \odot \tau_j. \tag{6}$$

Our masked model merging strategy has three main advantages: (1) effectively extracts information from the new task $\tau_j$; (2) removes interference present in $\tau_{seq}^{(j-1)}$ from the efficient merging process; and (3) ensures conflict-free parameter merging. Compared to existing methods, our approach is better suited for complex tasks and improves the robustness of model merging. Since the mask is useful, the next challenge is how to obtain an effective mask.

**Mask optimization should consider global tasks consensus.** Current localized-aware methods often focus solely on individual tasks, but effectiveness in a single task does not guarantee effectiveness after merging. Therefore, it is crucial to evaluate effectiveness from a global task perspective. In the context of serializing the merging task $j$, let $S_v$ denote the current set of visible tasks, which includes task $j$ and all preceding visible tasks. Leveraging the credible dataset obtained through the CB-EMS method, for each visible task $t_v \in S_v$, we can compute the empirical loss $\mathcal{L}_{t_v}(\theta)$. Therefore, the optimization objective for the mask is:

$$\min_M \sum_{t_v \in S_v} \mathcal{L}_{t_v}(\theta_{seq}^{(j)}) + \alpha \|M\|_1, \qquad (7)$$

where $\theta_{seq}^{(j)} = \theta_{pre} + \tau_{seq}^{(j)}$. As shown, our mask optimization takes into account all current visible tasks, allowing effective consensus to be extracted on a broader scale. In addition, we introduce the $L_1$ norm $\|M\|_1$ to promote sparsity in the mask, with $\alpha$ as the hyperparameter of the $L_1$ term.

Through the masked model merging and optimization processes, we successfully obtained the binary mask $M$ required for merging the task j, and completed the merging of task j, resulting in $\tau_{seq}^{(j)}$. However, effectively optimizing a binary mask is challenging. In practice, following (Panigrahi et al., 2023; He et al., 2024), we re-parametrize the binary mask as the sigmoid of a real-valued vector $R$, i.e., $M = \sigma(R)$ to facilitate the optimization process. Combining the above mask optimization method with the efficient-aware framework, we implement a serialized merging approach, as detailed in Algorithm 1. The final binary mask is obtained by rounding the sigmoid output, represented as $M^* = Round(\sigma(R))$, resulting in the next-step task vector:

$$\tau_{seq}^{(j)} = (1 - M^*) \odot \tau_{seq}^{(j-1)} + M^* \odot \tau_j. \qquad (8)$$

Finally, we obtain the merged model as follow:

$$\theta_{mtl}^{(final)} = \theta_{pre} + \tau_{seq}^{(final)}. \qquad (9)$$

# 5. Experiments

This section outlines our experimental setup and presents performance comparisons. Additionally, we provide a comprehensive analysis of the robustness and convergence of our CALM, as well as the effectiveness of CB-EMS.

---

**Algorithm 1** CALM

**Input:** Number of tasks $T$, pretrained model $\theta_{pre}$, fine-tuned models $\{\theta_{ft}^t\}_{t=1}^T$, task-specific unsupervised datasets $\{\mathcal{D}^t(\mathcal{X})\}_{t=1}^T$, efficient merging coefficient $\lambda$, regularization parameter $\alpha$.

1: **Step 1: Compute Task Vectors**
2: **for** each task $t$ from 1 to $T$ **do**
3:     Compute task vector $\tau_t = \theta_{ft}^t - \theta_{pre}$
4: **end for**
5: **Step 2: Class-Balanced EMS**
6: **for** each task $t$ from 1 to $T$ **do**
7:     Obtain class-balanced credible samples as Eq. 5;
8:     Assign pseudo-labels to create $\hat{\mathcal{D}}_{cb}^t(\mathcal{X}, \hat{\mathcal{Y}})$;
9: **end for**
10: **Step 3: Efficient-Aware**
11: Randomly select a few tasks as sequential tasks $\overline{S}$;
12: The rest form efficient merged tasks $S$;
13: Efficiently merge $S$ to obtain $\tau_{seq}^{(0)} = \lambda \sum_{t \in S} \tau_t$;
14: **Step 4: Consensus-Aware Mask Optimization**
15: visible tasks $S_v = S$, real-valued mask $R$;
16: **for** each task $j$ in $\overline{S}$ **do**
17:     Add task $j$ to $S_v$;
18:     Masked model merging:
19:     $\theta_{seq}^{(j-1)} = \theta_{pre} + (1 - \sigma(R)) \odot \tau_{seq}^{(j-1)} + \sigma(R) \odot \tau_j$;
20:     Masked model optimization:
21:     $R^* = \arg\min_R \sum_{t_v \in S_v} \mathcal{L}_{t_v}(\theta_{seq}^{(j-1)}) + \alpha\|\sigma(R)\|_1$;
22:     Binarization: $M^* = Round(\sigma(R^*))$;
23:     Update: $\tau_{seq}^{(j)} = (1 - M^*) \odot \tau_{seq}^{(j-1)} + M^* \odot \tau_j$;
24: **end for**
25: **Output:** Merged model $\theta_{mtl}^{(final)} = \theta_{pre} + \tau_{seq}^{(final)}$

---

## 5.1. Experimental Setup

**Baselines.** We compare CALM with eight representative model merging approaches. These baselines fall into three categories. The first includes traditional model merging methods that do not rely on task vectors, such as *Simple Averaging*, *Fisher Merging* (Matena & Raffel, 2022), and *RegMean* (Jin et al., 2023). The second category comprises global-aware methods, including *Task Arithmetic* (Ilharco et al., 2023), *AdaMerging* (Yang et al., 2024b) and *Pareto Merging* (Chen & Kwok, 2024). We evaluate two strategies for *AdaMerging*: task-wise (*TW AdaMerging*) and layer-wise (*LW AdaMerging*). The third category encompasses localized-aware methods, such as *TIES-Merging* (Yadav et al., 2023), *Consensus TA* (Wang et al., 2024), *Consensus TIES* (Wang et al., 2024) and *Localize-and-Stitch* (He et al., 2024). Additionally, we present three comparison results: the performance of pretrained models without merging (the lower bound), individual models and traditional MTL, which serve as the upper bound.

**Datasets.** Following previous work on generic datasets (Il-

| Method | SUN397 | Cars | RESISC45 | EuroSAT | SVHN | GTSRB | MNIST | DTD | Avg Acc |
|---|---|---|---|---|---|---|---|---|---|
| Pretrained | 62.3 | 59.7 | 60.7 | 45.5 | 31.4 | 32.6 | 48.5 | 43.8 | 48.0 |
| Individual | 75.3 | 77.7 | 96.1 | 99.7 | 97.5 | 98.7 | 99.7 | 79.4 | 90.5 |
| Traditional MTL | 73.9 | 74.4 | 93.9 | 98.2 | 95.8 | 98.9 | 99.5 | 77.9 | 88.9 |
| Weight Averaging | 65.3 | 63.4 | 71.4 | 71.7 | 64.2 | 52.8 | 87.5 | 50.1 | 65.8 |
| RegMean (Jin et al., 2023) | 65.3 | 63.5 | 75.6 | 78.6 | 78.1 | 67.4 | 93.7 | 52.0 | 71.8 |
| Fisher Merging (Matena & Raffel, 2022) | 68.6 | 69.2 | 70.7 | 66.4 | 72.9 | 51.1 | 87.9 | 59.9 | 68.3 |
| Task Arithmetic (Ilharco et al., 2023) | 55.2 | 54.9 | 66.7 | 78.9 | 80.2 | 69.7 | 97.3 | 50.4 | 69.1 |
| TW AdaMerging (Yang et al., 2024b) | 58.0 | 53.2 | 68.8 | 85.7 | 81.1 | 84.4 | 92.4 | 44.8 | 71.1 |
| LW AdaMerging (Yang et al., 2024b) | 64.5 | 68.1 | 79.2 | 93.8 | 87.0 | 91.9 | 97.5 | 59.1 | 80.1 |
| Pareto Merging (Chen & Kwok, 2024) | 71.4 | **74.9** | 87.0 | 97.1 | 92.0 | **96.8** | 98.2 | 61.1 | 84.8 |
| Ties-Merging (Yadav et al., 2023) | 59.8 | 58.6 | 70.7 | 79.7 | 86.2 | 72.1 | 98.3 | 54.2 | 72.4 |
| Consensus TA (Wang et al., 2024) | 63.9 | 64.1 | 75.5 | 79.4 | 81.6 | 69.9 | 98.0 | 55.1 | 73.7 |
| Consensus TIES (Wang et al., 2024) | 62.3 | 62.2 | 74.5 | 80.0 | 87.7 | 77.5 | 98.6 | 55.3 | 74.8 |
| Localize-and-Stitch (He et al., 2024) | 67.2 | 68.3 | 81.8 | 89.4 | 87.9 | 86.6 | 94.8 | 62.9 | 79.9 |
| **CALM (ours)** | **72.6** | 74.8 | **91.9** | **98.6** | **95.2** | 96.4 | **99.1** | **72.8** | **87.7** |

*Table 1.* Multi-task performance with CLIP ViT-B/32 architecture on eight vision classification tasks.

harco et al., 2023; Yang et al., 2024b; He et al., 2024), we evaluate the performance of CALM on 8 visual classification tasks and 12 natural language tasks. The eight visual classification datasets include SUN397 (Xiao et al., 2016), Cars (Krause et al., 2013), RESISC45 (Cheng et al., 2017), EuroSAT (Helber et al., 2019), SVHN (Netzer et al., 2011), GTSRB (Stallkamp et al., 2011), MNIST (LeCun, 1998), and DTD (Cimpoi et al., 2014). The natural language tasks consist of 12 GLUE tasks (Wang, 2018), including six single-sentence tasks: SST-2 (Socher et al., 2013), CR (Hu & Liu, 2004), MR (PaNgB, 2005), MPQA (Wiebe et al., 2005), TREC (Voorhees et al., 1999) and SUBJ (Lee & Pang, 2004), and six pairwise-sentence tasks: QNLI (Wang, 2018), SNLI (Bowman et al., 2015), MNLI (Williams et al., 2017), RTE (Wang, 2018), MRPC (Dagan et al., 2005) and QQP (Iyer et al., 2017). To avoid information leakage from the test set, 8 visual tasks use unsupervised training samples for optimization. For 12 NLP tasks, the validation set is used, while some datasets without a validation set use training samples. All data are unsupervised.

**Models and details.** For the visual experiments, we use the ViT-B/32 architecture from CLIP (Radford et al., 2021) as the pre-trained model, while for the NLP experiments, we use the RoBERTa-base (Liu, 2019) model, with pre-trained and fine-tuned models consistent with previous experiments (Yang et al., 2024b; He et al., 2024). The merging process randomly selects two tasks for sequential merging, while others apply task arithmetic (Ilharco et al., 2023) with a coefficient of 0.3. For the visual tasks, 90% of the credible samples are used, and for the NLP tasks, 80% are used. Each task is optimized for 100 iterations, with the regularization parameter $\lambda = 1$. For details, please refer to Appendix A.

### 5.2. Visual Experimental Results

Our experimental evaluations, conducted across various visual classification datasets, conclusively demonstrate the

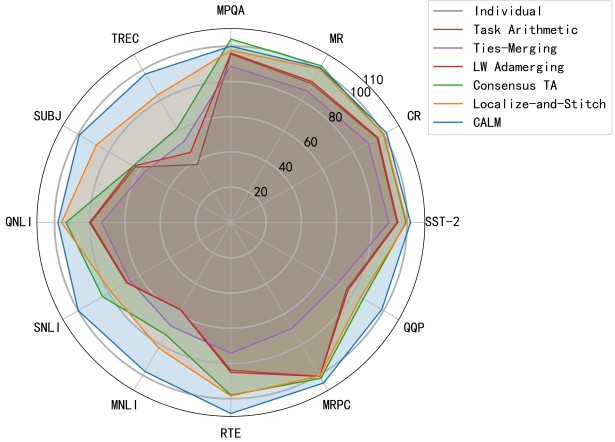

*Figure 3.* Multi-task performance with RoBERTa-base architecture on twelve NLP tasks, with results presented as absolute accuracy (%) compared to individual tasks.

superior performance of our CALM. Our analysis of the results presented in Table 1 leads to two primary insights:

**CALM significantly outperforms other baselines and approaches traditional MTL performance.** On most tasks, CALM demonstrates a 1%-3% performance improvement over existing global-aware and localized-aware model merging methods. Furthermore, when compared to traditional MTL, the performance of CALM is nearly on par with the theoretical upper bound achievable through retraining with data, a result attained with minimal computational cost.

**CALM ensures a more balanced performance across tasks.** While baseline methods excel on tasks like MNIST and SVHN, they suffer substantial performance declines (up to almost 20%) on tasks like RESISC45 and DTD when compared to the individual models. In comparison, CALM demonstrates minimal accuracy degradation on any task, achieving a more balanced performance. This highlights the reliability of CALM in multi-task model merging.

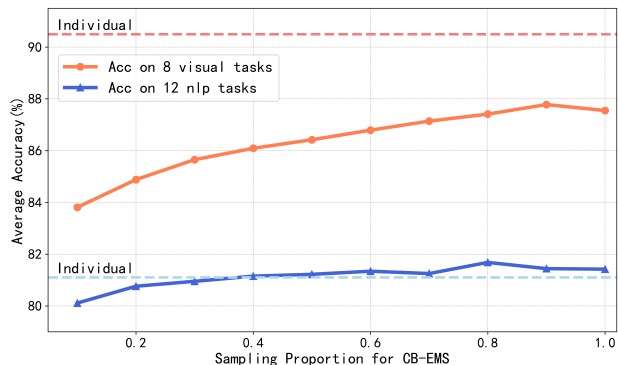

*Figure 4.* Variation of average accuracy with sampling proportion for CB-EMS across visual and NLP tasks.

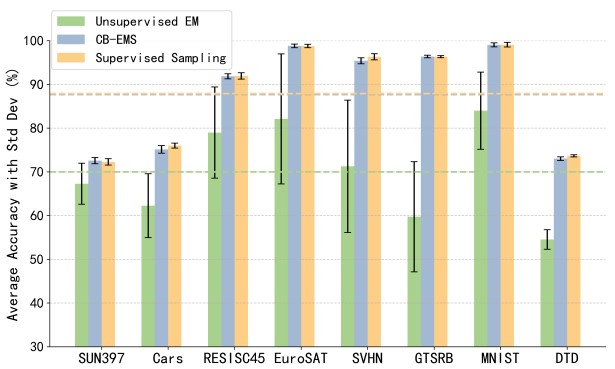

*Figure 5.* Average accuracy and standard deviation for entropy minimization sampling and its ablation methods.

### 5.3. NLP Experimental Results

In the NLP experiments, we compared several representative baselines with our method, as shown in Figure 3. The results in the figure are presented as absolute accuracy (%) compared to individual tasks, where an accuracy of 100% indicates that the performance of the merged model closely matches that of the individual fine-tuned models.

The experimental results demonstrate that our CALM approach achieves nearly the same performance as the individual models on most tasks, and even outperforms individual models on some tasks, such as RTE and MRPC, leading to an overall improvement in performance after merging. Additionally, CALM consistently performs well across all datasets, exhibiting superior consistency compared to the baselines, which show weaker performance on some tasks.

### 5.4. Analysis of CB-EMS

The experimental results have demonstrated that consensus-aware mask process successfully extracts effective information for model merging. Next, we analyze the influence of the sampling information on the results.

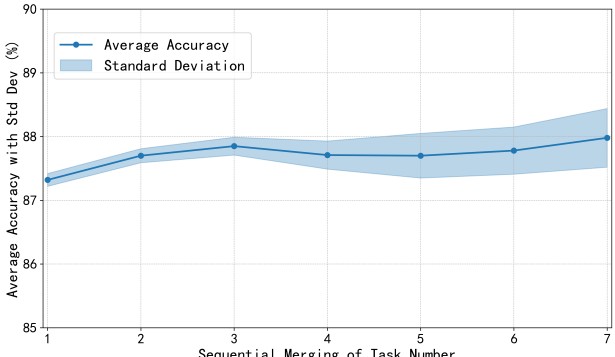

*Figure 6.* Average accuracy and standard deviation on visual tasks with different numbers of sequential merging tasks.

Figure 4 depicts the effect of sampling proportion on average accuracy across visual and NLP tasks. The accuracy initially increases and then decreases as the sampling proportion for CB-EMS rises, indicating that while more samples can improve performance, too many unreliable samples harm aggregation. The results demonstrate that a 30% sampling rate is sufficient for state-of-the-art(SOTA) performance in visual tasks, while a 50% sampling rate achieves individual task performance, also at SOTA, for NLP tasks.

Figure 5 compares CB-EMS with its ablation methods: the unsupervised EM method used by Adamerging and the supervised method with ground-truth labels. CB-EMS performs similarly to the supervised method and significantly outperforms the unsupervised EM method, suggesting that it retains a majority of the correct information. Furthermore, the standard deviation reveals that CB-EMS exhibits greater robustness, with a more stable merging process.

### 5.5. Analysis of Efficient-Aware Framework

In our sequential framework, two factors may introduce sensitivity into the experimental results: the order of sequential task merging and the number of tasks merged sequentially. In this section, we present experiments demonstrating that CALM exhibits substantial robustness in both aspects.

Figure 6 presents the average accuracy and variance of CALM on visual tasks with different numbers of sequential merging tasks $|\overline{S}|$. As the number of sequential tasks increases, computational complexity rises, and the average accuracy shows a slight increase, stabilizing around 87.7%.This indicates CALM's robustness to the number of sequential tasks, achieving effective model merging even with fewer tasks. The standard deviation also increases with more sequential tasks, introducing some randomness. However, the maximum standard deviation is about 0.4%, which does not significantly impact performance. Thus, CALM remains robust to the order of sequential tasks, with more stable performance when fewer tasks are used.

| Model Parameters | SUN397 | Cars | RESISC45 | EuroSAT | SVHN | GTSRB | MNIST | DTD | Avg Acc |
|---|---|---|---|---|---|---|---|---|---|
| $\theta_{pre}$ | 62.3 | 59.6 | 60.3 | 45.7 | 31.6 | 32.6 | 48.3 | 44.4 | 48.1 |
| $\theta_{pre} + \tau_1$ | 48.8 | 54.9 | 46.3 | 35.4 | 46.7 | 32.9 | 74.8 | 35.0 | 46.9 |
| $\theta_{pre} + (1 - M^*) \odot \tau_1$ | 60.4 | 64.8 | 69.8 | 85.6 | 76.5 | 68.6 | 89.5 | 54.0 | 71.2 |
| $\theta_{pre} + \tau_2$ | 75.3 | 77.7 | 96.1 | 99.7 | 97.5 | 98.7 | 99.7 | 79.4 | 90.5 |
| $\theta_{pre} + M^* \odot \tau_2$ | 69.9 | 69.5 | 80.0 | 88.1 | 72.7 | 71.7 | 90.4 | 59.7 | 75.3 |
| $\theta_{pre} + (1 - M^*) \odot \tau_1 + M^* \odot \tau_2$ | 71.9 | 73.9 | 88.3 | 96.9 | 93.4 | 93.9 | 98.0 | 69.0 | 85.7 |

*Table 2.* Multi-task performance under different model parameters. In contrast to Table 1, this experiment involves only one sequential merging task, where $\tau_2$ denotes the task vector for this task, $M^*$ represents its optimized binary mask, and $\tau_1$ corresponds to the task vector from the remaining seven tasks via task arithmetic.

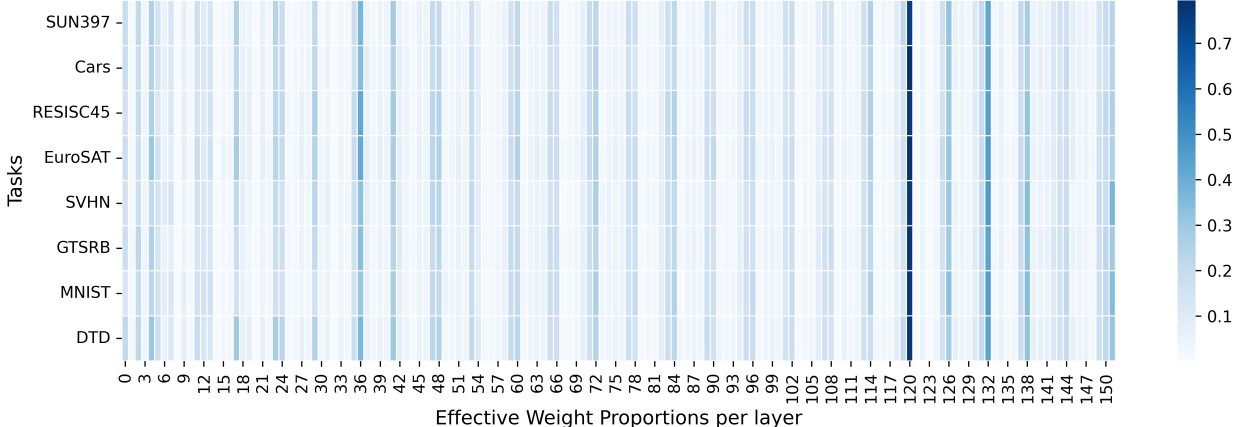

*Figure 7.* Effective weight proportions per layer across tasks visualized via consensus-aware mask.

### 5.6. Analysis of Consensus-Aware Mask

As shown in Table 2, we validate the effectiveness of the extracted consensus-aware mask. To intuitively demonstrate the mask's value for each task vector, we employ only one sequential merging task, while merging the remaining tasks efficiently via task arithmetic. Here $\theta_{pre}$ denotes the pre-trained model, $\tau_1$ is the task vector from the remaining seven tasks via task arithmetic, $\tau_2$ is the target task vector, and $M^*$ is the optimized binary mask. We evaluate the target task performance under five configurations.

As illustrated in the table, CALM effectively accomplishes model merging by precisely localizing 5% of critical local parameters with global consensus. Removing 5% of interference parameters from $\tau_1$ improves the average accuracy by nearly 25% and enhances target task adaptability, demonstrating its capability to accurately identify interference regions; restricting $\tau_2$ to 5% of masked parameters maintains strong performance, indicating minimal loss of task-relevant information and successful retention of critical transfer signals; simultaneously applying $\theta_{pre} + (1 - M^*) \odot \tau_1$ and $M^* \odot \tau_2$ achieves conflict-free model merging and outperforms independent adjustment strategies, verifying effective multi-task merging capability of CALM.

### 5.7. Visualization of Effective Weight Proportions

As illustrated in Figure 7, the effective weight proportions of each layer are depicted when each task serves as a serialized task, providing an intuitive representation of the importance of each layer. From the figure, it can be observed that there is a strong consistency across tasks: if a layer is important for one task, it is likely important for others as well. Additionally, some layers exhibit significant differences in weight proportions, highlighting the specificity between tasks.

## 6. Conclusion

In this study, we explore how to identify and extract effective information in model merging. By analyzing global-aware and local-aware methods, we propose that effective information is found in localized parameters aligned with global task consensus. Based on this insight, we introduce CALM, a new approach that uses reliable unsupervised data and an efficient-aware framework to identify and integrate parameters with global consensus. Extensive experiments show CALM's superiority over existing methods. This study provides a new perspective on model merging, focusing on aligning localized knowledge with global consensus, and offers a potential solution for multi-task and transfer learning.

## Impact Statement

This paper presents work whose goal is to advance the field of Machine Learning. There are many potential societal consequences of our work, none which we feel must be specifically highlighted here.

## Acknowledgements

This work is funded by the National Science and Technology Major Project (No. 2022ZD0114903) and the Natural Science Fundation of China (NSFC. No. 62476149). Sen Cui would like to acknowledge the financial support received from Shuimu Tsinghua scholar program. FL is supported by the Australian Research Council (ARC) with grant number DE240101089, LP240100101, DP230101540 and the NSF & CSIRO Responsible AI program with grant number 2303037.

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

# A. Experimental Details

## A.1. Datasets

Following the standard practices in the multi-task model merging field (Ilharco et al., 2023; Yadav et al., 2023; Yang et al., 2024b; He et al., 2024), we employ eight image classification datasets and twelve NLP datasets to evaluate the effectiveness of our proposed method. A detailed introduction to these eight datasets is provided below.

**Visual datasets:**

- **SUN397**(Xiao et al., 2016) is a large-scale scene recognition dataset with 397 categories and over 108,000 images, covering a diverse range of indoor and outdoor scenes to support scene-level attribute analysis.

- **Stanford Cars**(Krause et al., 2013) consists of 16,185 images from 196 car models, labeled by make, model, and year, aimed at fine-grained vehicle classification tasks.

- **RESISC45**(Cheng et al., 2017) is a remote sensing dataset with 31,500 images spanning 45 scene categories, representing different land-use types such as residential, industrial, and agricultural areas.

- **EuroSAT**(Helber et al., 2019) contains 27,000 satellite images from Sentinel-2 data, classified into 10 land-use categories, useful for applications in earth observation and environmental monitoring.

- **SVHN**(Netzer et al., 2011) includes over 600,000 images of house numbers from Google Street View, categorized into 10 classes (digits 0-9), with complex natural scene backgrounds, posing challenges for digit recognition.

- **GTSRB**(Stallkamp et al., 2011) contains more than 50,000 images of 43 types of traffic signs, used for evaluating traffic sign detection and classification, especially in autonomous driving systems.

- **MNIST**(LeCun, 1998) is a well-known dataset with 70,000 grayscale images of handwritten digits (0-9), serving as a benchmark for evaluating models in digit classification tasks.

- **DTD**(Cimpoi et al., 2014) consists of 5,640 images of 47 texture classes, each representing different texture types like "striped" or "dotted," commonly used for texture recognition and visual attribute understanding.

**NLP datasets:**

- **SST-2**(Socher et al., 2013) is a sentiment analysis dataset consisting of 67,349 sentences from movie reviews, labeled as either positive or negative, commonly used for binary sentiment classification tasks.

- **CR**(Hu & Liu, 2004) (Customer Review) is a sentiment classification dataset containing 1,000 product reviews, labeled as positive or negative, used to analyze customer opinions.

- **MR**(PaNgB, 2005) (Movie Review) contains 10,662 movie reviews, labeled as positive or negative, widely used for sentiment analysis in text classification tasks.

- **MPQA**(Wiebe et al., 2005) is a sentiment analysis dataset with 10,000 sentences from news articles, annotated for subjective and objective sentences, and used for polarity and subjectivity classification.

- **TREC**(Voorhees et al., 1999) is a question classification dataset with 6,000 labeled questions categorized into six broad topic categories, used for task-specific question classification.

- **SUBJ**(Lee & Pang, 2004) contains 10,000 subjective and objective sentences, labeled as subjective or objective, used for subjectivity classification tasks.

- **QNLI**(Wang, 2018) (Question Natural Language Inference) is a dataset with 104,000 pairs of questions and sentences, used to test if a sentence contains the answer to the given question (semantic entailment task).

- **SNLI**(Bowman et al., 2015) (Stanford Natural Language Inference) consists of 570,000 sentence pairs, labeled with entailment, contradiction, or neutral, used for natural language inference tasks.

- **MNLI**(Williams et al., 2017) (Multi-Genre Natural Language Inference) is a dataset containing 433,000 sentence pairs from various genres, used to classify if two sentences are in an entailment, contradiction, or neutral relationship.

- **RTE**(Wang, 2018) (Recognizing Textual Entailment) consists of 2,500 sentence pairs, focused on determining if one sentence logically follows from another, and is used for natural language inference tasks.

- **MRPC**(Dagan et al., 2005) (Microsoft Research Paraphrase Corpus) contains 3,600 sentence pairs, labeled as either paraphrases or non-paraphrases, used for paraphrase detection tasks.

- **QQP**(Iyer et al., 2017) (Quora Question Pairs) consists of 400,000 pairs of questions from Quora, labeled as duplicate or non-duplicate, commonly used for question pair similarity and paraphrase detection.

### A.2. Baselines

CALM is evaluated with respect to the following baseline approaches, which are introduced in further detail below.

- **Pretrained**. The model is pretrained on general-purpose datasets without any task-specific information. It serves as the lower bound for evaluating model merging methods, validating their effectiveness by demonstrating moderate task performance.

- **Individual**. These are models fine-tuned on each specific task based on the pretrained model. They serve both as foundational modules for model merging and as an upper bound to evaluate the performance loss during the merging process.

- **Traditional MTL**. This is a multi-task learning model trained directly on the combined training dataset. It serves as an upper bound for model merging methods, as these approaches aim to achieve similar performance without reusing training data.

- **Weight Averaging** averages the parameters of all individual models without considering parameter conflicts. It shows moderate improvement, slightly outperforming the pretrained model.

- **RegMean** (Jin et al., 2023) aims to ensure that the merged model remains close to individual fine-tuned models in L2 distance by using the covariance matrix of the training dataset to guide the merging process.

- **Fisher Merging** (Matena & Raffel, 2022) uses the Fisher information matrix, derived from training data, to guide the merging process. It treats the Fisher matrix as a measure of each parameter's importance in the model.

- **Task Arithmetic** (Ilharco et al., 2023) introduces the concept of task vectors, which are calculated as the difference between fine-tuned and pretrained models. It preserves task-specific information and merges task vectors using weighted aggregation to achieve significant performance gains.

- **TW AdaMerging** (Yang et al., 2024b) applies task-wise AdaMerging, where each task is assigned a merging weight. The merging weights are optimized using an unsupervised test dataset, enhancing the merging process over the Task Arithmetic method.

- **LW AdaMerging** (Yang et al., 2024b) utilizes layer-wise AdaMerging, providing finer granularity by assigning merging weights to individual layers. This approach effectively resolves parameter conflicts and achieves state-of-the-art performance on benchmark datasets.

- **Ties-Merging** (Yadav et al., 2023) focuses on merging parameters from a localized perspective. It uses strategies like Trim, Elect Sign, and Disjoint Merge to identify the most crucial parameters from each model while minimizing conflicts during the merging process.

- **Pareto Merging** (Chen & Kwok, 2024) formulates model merging as a multi-objective optimization problem, generating a diverse set of Pareto-optimal models in a single process. This approach outperforms existing methods by tailoring models to user-specific preferences.

- **Consensus TA** (Wang et al., 2024) proposes identifies the task supports given a collection of task vectors. This method, combined with Task Arithmetic, aggregates the selected task supports.

- **Consensus TIES** (Wang et al., 2024) proposes replacing the original Task Arithmetic module with Ties-Merging, achieving a strategy for selecting effective parameter points.

- **Localize-and-Stitch** (He et al., 2024) aims to identify important localized parameters by using partial training data for optimization. It strives to represent task information effectively with fewer parameters, compared to the TIES-Merging method.

### A.3. Implementation Details.

CALM employs an efficient-aware sequential merging approach, and we describe our experimental setup using the main experimental configuration as an example. First, we randomly select two tasks as the sequential tasks, while the remaining task vectors are merged using task arithmetic, formulated as $\tau_c = \lambda \sum_{i=1}^{n} \tau_i$, where $\lambda = 0.3$, following Task Arithmetic.

Next, we conduct sequential merging for the two randomly selected tasks. Using the class-balanced entropy minimization sampling method, We select 90% of the samples from the unlabeled samples of the visual task as the credible sample set, and 80% of the samples from the text task as the credible sample set. We initialize a real-valued mask $R$ of the same size as the model and set 1e-5 of the parameter points to be active. The mask $R$ is then iteratively trained with the credible sample set, which contains pseudo-labels, using a batch size of 128 and a learning rate of $1e7$—a large learning rate to ensure effective information feedback to the mask. For each iteration, only two batch of the reliable sample set per task is used, with a total of 100 iterations.

The trained real-valued mask $R$ is converted to a binary mask $M$ using the sigmoid function, and a new task vector is merged accordingly. Once all sequential tasks are merged, the final merged model is obtained. The same process is followed for other numbers of sequential tasks.

### A.4. Computing Resources

Part of the experiments is conducted on a local server with Ubuntu 16.04 system. It has two physical CPU chips which are Intel(R) Xeon(R) Gold 6248 CPU @ 2.50GHz with 20 cpu cores. The other experiments are conducted on a remote server. It has 8 GPUs which are GeForce RTX 3090.

## B. Class-Balanced Entropy Minimization Sampling Analysis

**EMS vs. CB-EMS: validation comparison.** Figure 8 shows the accuracy of pseudo-labels versus ground truth across eight tasks at different EMS sampling rates. As the entropy increases, the accuracy of the pseudo-labels decreases significantly, indicating a strong correlation between entropy and sample confidence. Therefore, compared to Adamerging, our selected samples exhibit greater robustness. Additionally, comparing (a) and (b) reveals that the accuracy of CB-EMS decreases more rapidly, This suggests that there are significant differences in entropy across classes, making EMS prone to class imbalance.

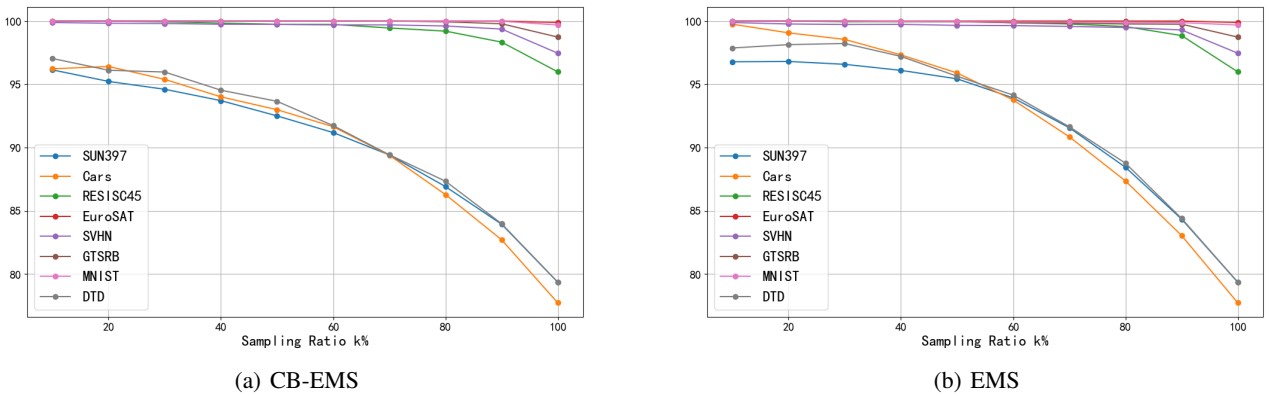

(a) CB-EMS          (b) EMS

*Figure 8.* Accuracy of EMS and CB-EMS pseudo-labels compared to ground truth across different sampling rates

## B.1. Entropy Imbalance Analysis

Figure 9 illustrates the boxplots of entropy for each class in the DTD and RESISC45 datasets. The boxplots visually represent the median entropy for each class, with the bottom and top of the box indicating the first quartile (Q1) and third quartile (Q3), respectively. The whiskers extend to the minimum and maximum values within the dataset. From these boxplots, it is evident that there are significant differences in entropy across classes—some classes have consistently low entropy, while others show predominantly higher values. Thus, employing the class-balanced entropy minimization sampling method is justified, as it helps prevent sample selection from being dominated by only a few classes.

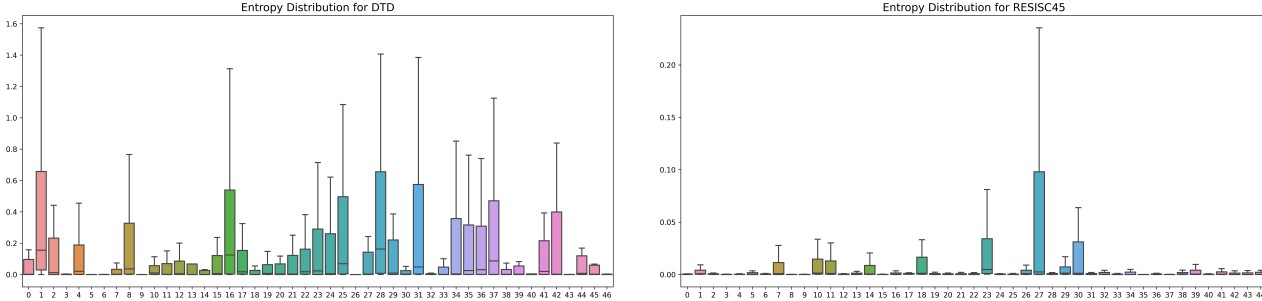

*Figure 9.* Binned boxplot of entropy distribution for DTD and RESISC45 datasets

## B.2. Entropy Minimization Reliability Analysis

This subsection aims to demonstrate that directly using the entropy minimization method is insufficient due to the unreliability of the resulting entropy. As illustrated in Figure 10, both the adamerging and CALM methods start with an initial model equivalent to task arithmetic. Therefore, we visualize the confusion matrices of this initial model for the DTD and RESISC45 tasks. From the figure, it is evident that the initial model yields suboptimal predictions for these tasks, with many classes being almost entirely misclassified. Consequently, the entropy estimated from these samples points in an incorrect direction.

Using an entropy minimization approach in this context could further enhance the credibility of these incorrect entropy values, causing these erroneous signals to negatively impact model merging. In contrast, the CALM method alleviates the impact of these misleading entropies by first sampling a reliable dataset, thus mitigating the harm. Moreover, CALM sets pseudo-labels to ensure that entropy information remains unaffected during model merging, ultimately leading to more reliable outcomes.

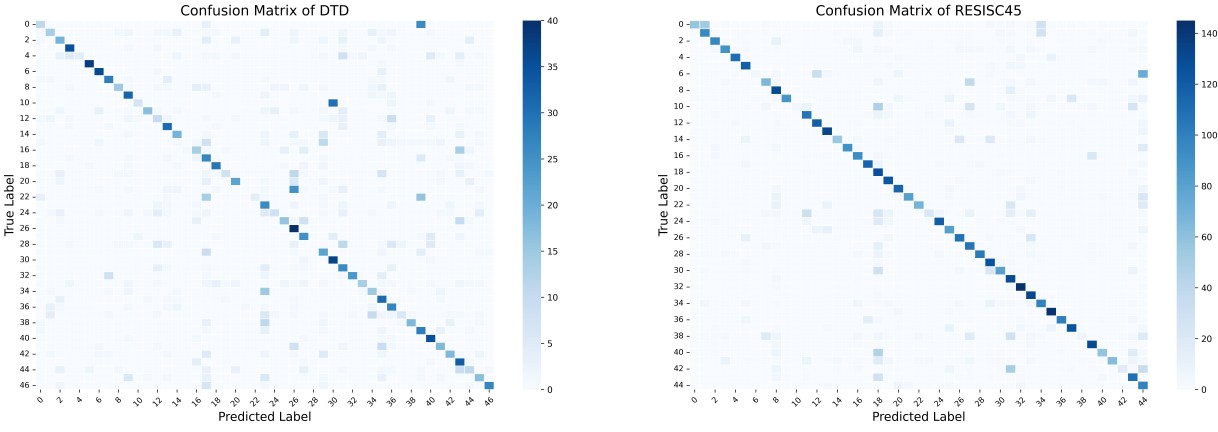

*Figure 10.* Confusion matrix of initial model predictions on DTD and RESISC45 tasks

# C. Supplementary Experiment

## C.1. Merging Strategy Analysis

CALM introduces a consensus-aware, conflict-free merging strategy, and we explore the effectiveness of this merging approach below. Given a binary mask $M$, CALM's merging method is defined as $\tau_{mtl} = (1 - M) \odot \tau_{seq} + M \odot \tau_j$. The advantage of this approach is that all tasks' task vectors participate in mask training, while ensuring that the parameters of the new task vector do not conflict with those of the original task vectors. In our ablation study, we also investigate the performance of merging strategies with only $(1 - M)$ and only $M$.

The experimental results are shown in Table 3. It can be observed that the merging strategies using only $M$ or only $(1 - M)$ exhibit distinct characteristics. The strategy with only $M$ maintains the state of the original tasks better, as it does not affect the information of the existing tasks, while the new task is impacted due to parameter interference. In contrast, the strategy using only $(1 - M)$ retains the new task information well but results in a decrease in the original tasks' performance due to information loss and parameter conflicts. By comparison, CALM achieves a better balance by combining both approaches, leading to superior performance overall.

| Method | Merging Strategy | SUN397 | Cars | RESISC45 | EuroSAT | SVHN | GTSRB | MNIST | DTD | Avg Acc |
|---|---|---|---|---|---|---|---|---|---|---|
| LW AdaMerging | global | 64.5 | 68.1 | 79.2 | 93.8 | 87.0 | 91.9 | 97.5 | 59.1 | 80.1 |
| Localize-and-Stitch | local | 67.2 | 68.3 | 81.8 | 89.4 | 87.9 | 86.6 | 94.8 | 62.9 | 79.9 |
| CALM | only $M$ | 72.9 | 73.4 | 89.6 | 98.1 | 95.1 | 95.1 | 98.8 | 69.9 | 86.6 |
| CALM | only $1 - M$ | 69.0 | 70.6 | 87.2 | 96.6 | 93.0 | 93.3 | 98.0 | 77.5 | 85.7 |
| CALM | both (ours) | 72.6 | 74.8 | 91.9 | 98.6 | 95.2 | 96.4 | 99.1 | 72.8 | 87.7 |

*Table 3.* Multi-task performance on eight vision classification tasks under different merging strategies.

## C.2. Learning Rate Analysis

Table 4 investigates the influence of learning rate on our proposed method. In the main experiment, the learning rate is set to 1e7. We further evaluate the effects of learning rates 5e6 and 2e7, and compare the results with those obtained by the localize-and-stitch method under identical learning rates. The results indicate that the impact of different learning rates on CALM is negligible, with each dataset showing consistent performance fluctuations. This demonstrates the stability of our method across varying learning rates, underscoring the robustness of CALM.

| Method | Learning Rate | SUN397 | Cars | RESISC45 | EuroSAT | SVHN | GTSRB | MNIST | DTD | Avg Acc |
|---|---|---|---|---|---|---|---|---|---|---|
| Localize-and-Stitch | 1e7 | 67.2 | 68.3 | 81.8 | 89.4 | 87.9 | 86.6 | 94.8 | 62.9 | 79.9 |
| CALM | 5e6 | 72.4 | 74.6 | 91.8 | 98.7 | 95.2 | 96.4 | 98.9 | 71.6 | 87.5 |
| CALM | 2e7 | 72.2 | 74.3 | 92.1 | 98.7 | 95.3 | 96.2 | 99.1 | 73.2 | 87.6 |
| CALM | 1e7 | 72.6 | 74.8 | 91.9 | 98.6 | 95.2 | 96.4 | 99.1 | 72.8 | 87.7 |

*Table 4.* Multi-task performance on eight vision classification tasks under different learning rates.

## C.3. Regularization Coefficient Analysis

Table 5 investigates the impact of the L1 norm regularization coefficient on the performance of CALM. This coefficient controls the extent to which regularization influences the final result, with larger coefficients encouraging a sparser mask. The results indicate that the regularization coefficient has minimal effect on overall performance. While larger coefficients may lead to a slight decline in CALM's performance, CALM generally maintains stability across different values of the regularization coefficient, demonstrating its robustness.

| Method | SUN397 | Cars | RESISC45 | EuroSAT | SVHN | GTSRB | MNIST | DTD | Avg Acc |
|---|---|---|---|---|---|---|---|---|---|
| Pretrained | 66.8 | 77.7 | 71.0 | 59.9 | 58.4 | 50.5 | 76.3 | 55.3 | 64.5 |
| Individual | 82.3 | 92.4 | 97.4 | 100 | 98.1 | 99.2 | 99.7 | 84.1 | 94.2 |
| Traditional MTL | 80.8 | 90.6 | 96.3 | 96.3 | 97.6 | 99.1 | 99.6 | 84.4 | 93.5 |
| Weight Averaging | 72.1 | 81.6 | 82.6 | 91.9 | 78.2 | 70.7 | 97.1 | 62.8 | 79.6 |
| RegMean (Jin et al., 2023) | 73.3 | 81.8 | 86.1 | 97.0 | 88.0 | 84.2 | 98.5 | 60.8 | 83.7 |
| Fisher Merging (Matena & Raffel, 2022) | 69.2 | 88.6 | 87.5 | 93.5 | 80.6 | 74.8 | 93.3 | 70.0 | 82.2 |
| Task Arithmetic (Ilharco et al., 2023) | 73.9 | 82.1 | 86.6 | 94.1 | 87.9 | 86.7 | 98.9 | 65.6 | 84.5 |
| Ties-Merging (Yadav et al., 2023) | 76.5 | 85.0 | 89.3 | 95.7 | 90.3 | 83.3 | 99.0 | 68.8 | 86.0 |
| AdaMerging (Yang et al., 2024b) | 79.0 | 90.3 | 90.8 | 96.2 | 93.4 | 98.0 | 99.0 | **79.9** | 90.8 |
| Pareto Merging (Chen & Kwok, 2024) | **82.5** | **91.4** | 92.0 | 98.5 | 95.4 | **98.6** | 99.0 | 78.3 | 92.0 |
| **CALM (ours)** | 80.2 | 90.8 | **94.6** | **98.8** | **96.2** | 97.5 | **99.3** | 79.3 | **92.1** |

*Table 6.* Multi-task performance with CLIP ViT-L/14 architecture on eight vision classification tasks.

| Method | Coefficient | SUN397 | Cars | RESISC45 | EuroSAT | SVHN | GTSRB | MNIST | DTD | Avg Acc |
|---|---|---|---|---|---|---|---|---|---|---|
| Localize-and-Stitch | 1 | 67.2 | 68.3 | 81.8 | 89.4 | 87.9 | 86.6 | 94.8 | 62.9 | 79.9 |
| CALM | 0.5 | 72.9 | 75.2 | 92.3 | 99.0 | 95.3 | 96.3 | 99.1 | 72.9 | 87.8 |
| CALM | 2 | 72.5 | 74.8 | 91.8 | 98.7 | 95.3 | 96.3 | 99.1 | 72.1 | 87.6 |
| CALM | 1e7 | 72.6 | 74.8 | 91.9 | 98.6 | 95.2 | 96.4 | 99.1 | 72.8 | 87.7 |

*Table 5.* Multi-task performance on eight vision classification tasks under different regularization coefficients.

## C.4. ViT-L/14 Experimental Results

Following the experimental settings used in previous work on model merging (Ilharco et al., 2023; Yadav et al., 2023; Yang et al., 2024b), we extend our results on the same datasets and methods using the CLIP ViT-L/14 architecture, as shown in Table 6. Given the larger scale of this model, the fine-tuned task information is well preserved, allowing even simpler approaches like weight averaging to achieve relatively strong model merging performance. Methods like adaMerging are already approaching the upper limit of model merging for this architecture, as demonstrated by individual and traditional MTL baselines, leaving limited room for further improvement.

Nonetheless, our approach still achieves state-of-the-art performance, improving the average accuracy by 0.1% over Pareto Merging, with only a 1.5% gap from the theoretical upper bound. This consistent, strong performance highlights both the effectiveness and stability of our method. Upon further analysis of individual datasets, it is evident that some datasets, such as Cars and MNIST, have nearly reached their theoretical upper bound, where CALM shows similar results to baselines. However, for datasets with greater room for improvement, such as RESISC45 and SVHN, CALM demonstrates significant performance gains. It is noteworthy that our method shows a slight decrease in performance on the DTD dataset, likely due to the fact that extracting reliable information for DTD as a serialized task requires more iterations. Increasing the iteration count presents a potential opportunity for further improvement in our approach.

## C.5. Convergence Analysis

We analyze the convergence of the CALM iterative process. Figure 11 exhibits the task sequence for the CALM($|\bar{S}| = 2$) setup, where MNIST and DTD datasets are sequentially merged into the model. Each task undergoes 100 iterations, with performance evaluated across all tasks every 10 iterations. The red curve represents the average accuracy at each evaluation point. Initially, the model is initialized using task arithmetic (Ilharco et al., 2023) over the first six tasks, resulting in the initial merged model. After 100 iterations, DTD is merged as the sequential task.

From Figure 11, we observe that CALM achieves rapid convergence, with performance improving significantly after just 10 iterations, almost reaching a stable state. The model exhibits high stability, with a consistent and smooth increase in accuracy. When DTD is added, the model experiences a slight perturbation, but previous performance is largely maintained. As training continues, the model's predictive ability improves further, indicating that CALM converges quickly and remains resilient to the integration of new tasks.

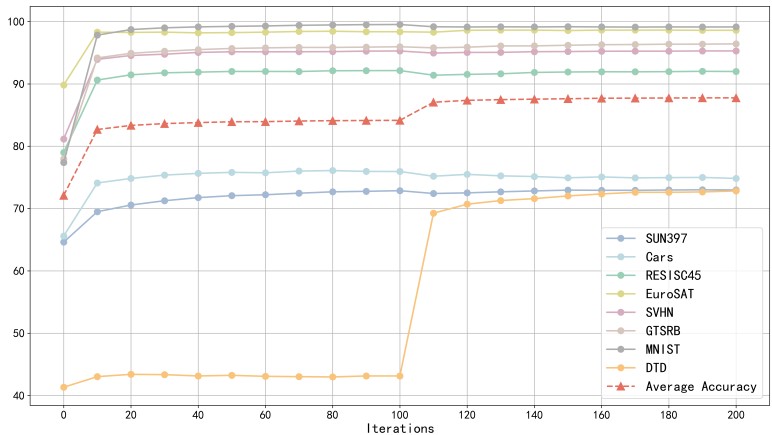

*Figure 11.* Accuracy and average accuracy curves over iterations for each dataset.

# D. Binary Mask Analysis

In the preceding sections, we have analyzed the effectiveness and robustness of the CALM method in model merging from a macro perspective. Below, we shift our focus to the binary mask representation to examine the specific characteristics of the parameter points identified by CALM.

### D.1. Binary Mask Activation Trend

We selected the CALM($|S| = 7, |\overline{S}| = 1$) configuration to explore the characteristics of the binary mask, which is generated for the final task to extract effective parameters while removing the influence of the previous seven merged tasks. Figure 12 illustrates the variation in the proportion of parameters set to 1 in the binary mask throughout the iterations. As shown in the figure, CALM identifies approximately 3.5%-4% of parameters as effective, indicating a high degree of redundancy in the model parameters. Initially, the mask randomly selects only 1e-5 of parameters to be set to 1, and this ratio increases gradually during iterations, demonstrating consistency across different datasets.

Additionally, an interesting phenomenon is observed: for more challenging tasks, such as SUN397, Cars, and DTD, the proportion of selected effective parameters is relatively higher, whereas for simpler tasks, like MNIST and SVHN, the ratio is lower. This aligns with our expectations, as more difficult tasks require more parameters to adequately represent their complexity.

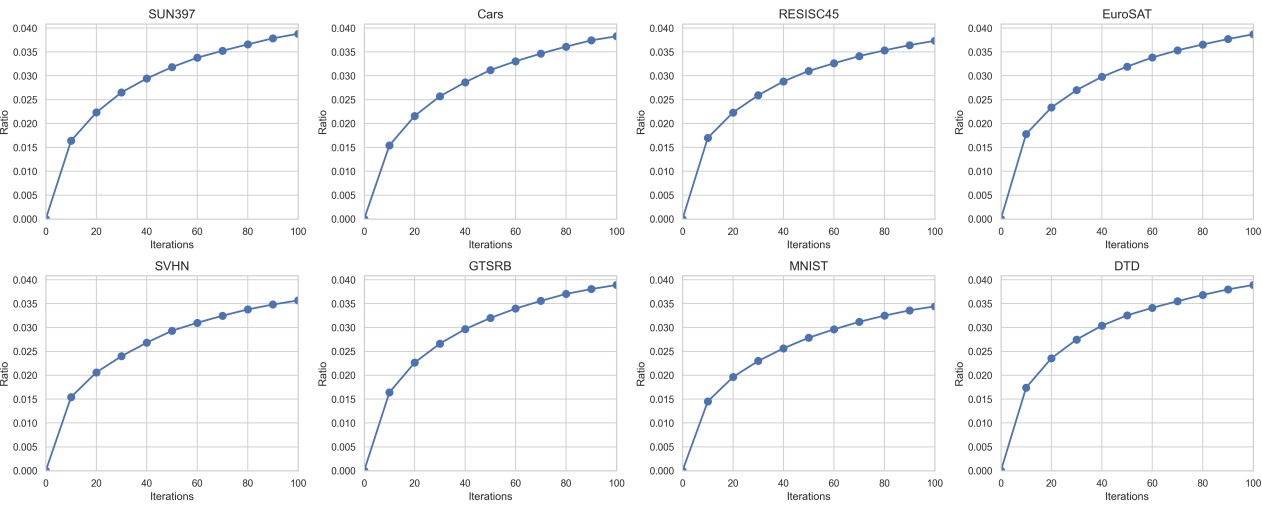

*Figure 12.* Effective weight proportions across iterations.

## D.2. Magnitude and Importance Correlation Analysis

For a long time, many studies have intuitively assumed that parameters with larger magnitudes in the task vector are more important, providing theoretical guidance for some data-free approaches, such as ties-merging (Yadav et al., 2023). In CALM, we identified better local parameter points with the assistance of unsupervised samples, which allows us to further analyze the relationship between parameter magnitude and its importance within the task vector. As shown in Table 7, the proportion of effective parameters with mask value 1 that rank within the top k% in terms of magnitude is presented for each task. If there were no correlation, the proportion in the table should approximately equal k. However, the results indicate that the magnitude of parameters in the task vector is correlated with their importance—parameters with larger magnitudes are more likely to be important. However, this does not imply that parameters with the highest magnitudes are always critical; rather, magnitude is merely one of several factors influencing importance. Therefore, methods like ties-merging do not necessarily identify the most significant parameters, but require sample information to more accurately determine these key parameters.

| Magnitude Selection | SUN397 | Cars | RESISC45 | EuroSAT | SVHN | GTSRB | MNIST | DTD |
|---|---|---|---|---|---|---|---|---|
| Top 1% | 4.9 | 4.6 | 3.5 | 2.3 | 3.6 | 4.9 | 3.9 | 3.9 |
| Top 5% | 17.5 | 16.0 | 14.1 | 10.7 | 14.1 | 16.4 | 14.6 | 14.9 |
| Top 10% | 29.2 | 26.8 | 24.6 | 19.9 | 24.6 | 27.0 | 24.8 | 25.5 |
| Top 20% | 46.6 | 43.3 | 41.0 | 35.0 | 40.9 | 42.8 | 40.4 | 41.6 |
| Top 50% | 76.8 | 73.9 | 72.0 | 66.3 | 71.4 | 71.9 | 70.1 | 71.9 |

*Table 7.* Proportion of effective parameters within the mask ranked in the top k% of magnitude across eight vision classification tasks.

# E. Time Cost Analysis

Benefiting from the efficient-aware framework, despite employing a sequential merging approach, we still achieve remarkable model merging performance with relatively low time cost. Analyzing from a convergence perspective, as demonstrated in Section 5.5, CALM achieves nearly a 10% improvement within just 10 iterations, almost reaching convergence. This process takes only about 4 minutes on a single GeForce RTX 3090, which is significantly faster compared to other model merging methods that rely on sample-based optimizations. After 10 iterations, our method converges gradually to the optimal solution with high stability. This feature allows for flexible adjustment of the number of iterations depending on computational resources and time constraints in practical scenarios.

Below, we compare the overall runtime of model merging optimization methods that require data. As shown in Table 8, we report the runtime costs for different numbers of serialized tasks. The localize-and-stitch method optimizes for a single task with 80 iterations, resulting in relatively low time consumption. By randomly selecting one serialized task for merging, we achieve a comparable runtime, while our performance far surpasses that of localize-and-stitch. As the number of serialized tasks increases, the runtime for CALM exhibits a sub-linear growth trend. This is due to the fact that each serialization step only considers previously seen tasks. Even in the most complex scenario, our runtime remains lower than that of adamerging, highlighting the efficiency of our approach.

| Method | Run-time Consumption | Iterations |
|---|---|---|
| Adamerging | 210.23min | 500 |
| CALM($|\overline{S}| = 7$) | 163.47min | 700 |
| CALM($|\overline{S}| = 2$) | 71.24min | 200 |
| CALM($|\overline{S}| = 1$) | 36.42min | 100 |
| Localize-and-Stitch | 31.76min | 80 |

*Table 8.* Run-time consumption of across different methods.

# F. Privacy Discussion

Model merging offers superior privacy protection compared to traditional multi-task learning, as it does not require retraining with the original training datasets. The information we use is identical to adamerging, relying solely on unsupervised test data, ensuring there is no risk of data leakage or privacy breaches. Moreover, CALM provides greater security and flexibility

compared to methods that either extract information from training data or directly use the training dataset. Specifically, CALM does not require direct interaction with data providers, which reduces the complexity of data access and minimizes privacy risks. Furthermore, unsupervised test samples may originate from diverse environments or distributed sources, making the approach adaptable for decentralized applications. Importantly, there is no need for additional manual labeling, enhancing both scalability and ease of implementation.

