# OpenReview forum: "CALM: Consensus-Aware Localized Merging for Multi-Task Learning"
_ICML.cc/2025/Conference — ICML 2025 poster_

### Official Review · Reviewer_hiCE · 2025-03-10

**Overall Recommendation:** 4

**Summary:**

The authors introduce a novel model merging approach called CALM to address multi-task learning integration. The core idea involves identifying localized parameters aligned with global task consensus through three key components:
1. Class-Balanced Entropy Minimization Sampling (CB-EMS): A method to extract reliable unsupervised datasets while preserving class balance.
2. Efficient-Aware Framework: A sequential merging strategy that reduces computational complexity.
3. Consensus-Aware Mask Optimization: A binary mask mechanism to extract effective localized parameters, where global consensus is optimized using CB-EMS datasets.
Experiments across diverse vision and language multi-task scenarios validate the method’s robustness and effectiveness.

**Claims And Evidence:**

1. From the insight, the author proposes that local parameters with global task consensus are the effective information in model merging. The author provides explanations and experiments, and I hope the author will further explore this global task consensus, which will be elaborated later.
2. The author proposes using Class-Balanced Entropy Minimization Sampling to build reliable unsupervised datasets. This part has sufficient proof.
3. The author proposes an Efficient-Aware Framework to achieve more efficient model merging. The explanation here is adequate but requires more experimental support.
4. The author introduces the Consensus-Aware Mask Optimization method to extract local parameters with global consensus. I agree with the value of masks in model merging, and the experimental results prove its effectiveness, since the improvement mainly comes from this method.

**Essential References Not Discussed:**

Maybe [1] can be added into discussion.
[1] Model breadcrumbs: Scaling multi-task model merging with sparse masks.

**Experimental Designs Or Analyses:**

The authors followed existing experimental setups, testing on standard visual and NLP benchmark datasets, and achieved significant results. However, some existing model merging methods have expanded to more tasks—further exploration on these benchmarks is warranted. The analysis of CB-EMS and Efficient-Aware is complete, but the authors should add more analysis for the Consensus-Aware Mask Optimization module.

**Methods And Evaluation Criteria:**

The authors offer a detailed explanation in the method section, including effective figures and algorithm workflows. CB-EMS originates from the classic Entropy Minimization Sampling approach and can be seen as an improvement on the Adamerging entropy.
The Efficient-Aware Framework is a serialized framework, with the final Mask Optimization stage aimed at refining task-specific binary masks applicable to all tasks. The method itself has no critical flaws, but the written of Mask Optimization and its corresponding algorithm do not precisely align. I believe the evaluation criteria are appropriately set.

**Other Comments Or Suggestions:**

Ensure alignment between the main text and algorithm diagrams, or provide more prominent explanations.

**Other Strengths And Weaknesses:**

## Strengths
1. This paper presents clear and impactful insight, with strong introductory explanations that offer valuable insights for model merging.
2. The introduction of EMS and sequentialized merging is innovative. While not entirely novel, to my knowledge, these ideas are the first proposed in the context of model merging, marking an exploratory contribution.
3. The proposed global task consensus is conceptually related to "task-shared information" but distinct: global consensus focuses on non-conflicting parameters, whereas task-shared information emphasizes parameter fusion. This distinction is thought-provoking.
## Weaknesses
However, I still have some questions that, if addressed, would significantly strengthen the work:
1. The Efficient-Aware Framework is innovative and proven effective. However, whether this framework applies to existing model merging methods requires further investigation.
2. The authors should provide a clearer explanation and brief exploration of how global task consensus is captured through Mask Optimization in practice.
3. Some existing model merging methods have expanded to more tasks， such as 14- 20 visual tasks—further testing on these benchmarks would solidify the claims.
4. Additional analysis of the Consensus-Aware Mask Optimization module is necessary such as the other two modules.

**Questions For Authors:**

My questions have been comprehensively outlined in the Weaknesses.

**Relation To Broader Scientific Literature:**

From a broader scientific perspective, model merging is similar to fields like federated learning and model fusion, but differs in that its multi-task training needs to be based on the same pretrained model. Existing methods fully leverage this characteristic by proposing task vectors [1], and how to optimally utilize these task vectors has become a key focus across approaches. The optimization strategy proposed by the authors – using an unsupervised test set to identify more effective parameter points (working points) for task vectors – builds upon existing research [2,3,4] and applies the optimization strategy to local parameters.

[1]Ilharco, G., Ribeiro, M. T., Wortsman, M., Schmidt, L., Hajishirzi, H., and Farhadi, A. Editing models with task arithmetic.

[2]Yadav, P., Tam, D., Choshen, L., Raffel, C. A., and Bansal, M. Ties-merging: Resolving interference when merging models.

[3]Yang, E., Wang, Z., Shen, L., Liu, S., Guo, G., Wang, X.,and Tao, D. Adamerging: Adaptive model merging for multi-task learning.

[4]He, Y., Hu, Y., Lin, Y., Zhang, T., and Zhao, H. Localize-and-stitch: Efficient model merging via sparse task arithmetic.

**Theoretical Claims:**

I have reviewed the formulas and equations in the method section. The paper's theoretical claims focus on explaining the method workflow, with no clear issues.

---

> ### Author Rebuttal · Authors · 2025-03-30
>
> ***Question 1: Does the Efficient-Aware Framework apply to existing model merging methods?***
>
> ***Answer:*** Thanks for the inspiring question.
>
> The Efficient-Aware Framework (EAF) introduces a novel serialized merging approach for model merging, which impacts existing methods as follows:
>
> - **For task quantity/order-insensitive methods**: EAF does not directly affect merging results and can be naturally applied to methods like Task Arithmetic and Localize-and-Stitch. These approaches typically disregard global task information during merging.
>
> - **For task quantity/order-sensitive methods**: Existing techniques (e.g., Ties-Merging, Adamerging) can adopt EAF through pairwise merging substitution. However, their performance will significantly degrade due to EAF's inability to comprehensively integrate multi-task information. Experimental validation appears in **Reviewer oEok Question 2**. This limitation further constrains their scalability for new tasks.
>
> ---
>
> ***Question 2: How is global task consensus captured through Mask Optimization in practice?***
>
> ***Answer:*** Thank you for the valuable question. We would like to explain it as follows:
>
> - **Theoretical perspective**:  The Mask Optimization process extracts gradient directions beneficial for global tasks, aligning with the global task consensus principle. Detailed theoretical analysis is provided in **Reviewer y6gA Question 1**.
>
> - **Empirical perspective**: Mask Optimization effectively compresses task-relevant information while eliminating inter-task interference. Experimental results are available in **Reviewer Rsjn Question 3**.
>
> ---
>
> ***Question 3: Additional experiments on extensive visual tasks.***
>
> ***Answer:*** Thanks for the constructive comments.
>
> We supplement six new datasets (**CIFAR100, Flowers102, OxfordIIITPet, STL10, KMNIST, FashionMNIST**), expanding the vision tasks to 14 in total, to validate CALM's stability under increased task diversity. Results are as follows.
>
> |Method|SUN397|Cars|RESISC45|EuroSAT|SVHN|GTSRB|MNIST|DTD|
> |-|-|-|-|-|-|-|-|-|
> |CALM ($\|S\|=13, \|\overline{S}\|=1$)|69.6|68.6|89.2|97.7|93.6|93.2|98.2|67.5|
> |CALM ($\|S\|=12, \|\overline{S}\|=2$)|71.4|73.4|91.0|98.3|94.6|95.2|98.4|70.4|
> |CALM ($\|S\|=11, \|\overline{S}\|=3$)|72.0|74.1|91.4|98.1|94.7|95.4|98.5|71.0|
> |CALM ($\|S\|=10, \|\overline{S}\|=4$)|72.3|74.0|91.5|98.1|94.9|95.5|98.6|71.7|
> |**Method**|**CIFAR100**|**Flowers102**|**OxfordIIITPet**|**STL10**|**KMNIST**|**FashionMNIST**|**Avg Acc**||
> |CALM ($\|S\|=13, \|\overline{S}\|=1$)|79.4|83.4|91.5|97.6|84.4|90.7|**86.0**||
> |CALM ($\|S\|=12, \|\overline{S}\|=2$)|81.9|85.0|91.4|97.9|87.1|90.8|**87.6**||
> |CALM ($\|S\|=11, \|\overline{S}\|=3$)|82.6|84.8|91.6|97.9|85.6|90.6|**87.7**||
> |CALM ($\|S\|=10, \|\overline{S}\|=4$)|83.2|84.7|91.8|97.9|86.2|90.7|**87.9**||
>
> **Key Findings from Experiments on 14 Visual Tasks:**
>
> - **Scalability:** CALM achieves robust performance (average accuracy ≈87% for the original 8 tasks) through sequential merging of a minimal task subset, demonstrating sustained effectiveness under task scaling.
> - **Flexibility:** Superiority over existing methods is attained even with limited merging steps, highlighting its efficiency in resource-constrained scenarios.
> - **Robustness:** The ordering of sequential merging tasks shows negligible impact on final performance, confirming task-agnostic stability.
>
> ---
>
> ***Question 4: Additional analysis of the Consensus-Aware Mask Optimization module.***
>
> ***Answer:*** Thanks for the practical comments.
>
> In the review, we provided a more detailed analysis of the Consensus-Aware Mask Optimization module, including:
>
> - **The definition of global consensus**: refer to **Reviewer Rsjn Question 1**;
> - **Experimental analysis of the binary mask**: refer to **Reviewer Rsjn Question 3**;
> - **Ablation Study of Consensus-Aware Mask Optimization**: refer to **Reviewer oEok Question 3**.
>
> ---
>
> ***Question 5: Essential References Discussion.***
>
> ***Answer:*** Thank you for your constructive advice.
>
> **Comparison between CALM and ***Model Breadcrumbs: Scaling Multi-Task Model Merging with Sparse Masks:*****
>
> - **Commonality:**
> Both methods leverage task vectors and acknowledge the significance of local parameters for multi-task model merging.
> - **Divergence:**
> Model Breadcrumbs filters anomalous parameter values based on local prior characteristics (e.g., magnitude distribution thresholds), whereas CALM dynamically extracts task-agnostic knowledge to identify optimal local parameters that align with global task consensus.
> - **Advantage of CALM:**
> CALM more effectively resolves conflicts among local parameters by integrating cross-task knowledge. The consensus-aware optimization ensures that selected parameters exhibit enhanced reliability and generalizability across diverse tasks, thereby offering a more principled and robust solution.

---

> > ### Comment · Reviewer_hiCE · 2025-04-04
> >
> > Thank you to the authors for the detailed and thoughtful response. After carefully reviewing the rebuttal, I find that it satisfactorily addresses my main concerns and provides additional clarity on several key aspects of the work. Specifically, the clarification of the Efficient-Aware Framework’s current scope and its potential applicability to other model merging methods is appreciated. The expanded analysis of the CAMO module also offers a deeper understanding of its contribution to the overall design. Accordingly, I have raised my score to 4.

---

### Official Review · Reviewer_oEok · 2025-03-10

**Overall Recommendation:** 4

**Summary:**

The paper introduces a test-time adaption method named CALM, which optimizes an equally-sized mask on pre-fine-tuned task-specific models through a reliable unsupervised dataset. The mask aims to extract locally shared parameters with global task consensus, offering a new perspective for model merging by exploring how to identify more effective local parameters. Furthermore, CALM implements a serialized merging framework that achieves remarkable efficiency in scenarios requiring individual merging of new tasks.

**Claims And Evidence:**

The paper focuses on addressing the question: "How to identify and extract effective information in model merging to enhance performance on all tasks?" The authors propose that effective information can be represented through localized parameters and aligned with a global task consensus.At first glance, this claim appears reasonable. The authors support their arguments through two primary approaches: first, by analyzing the limitations of existing global and localized model merging methods; second, by experimentally demonstrating the feasibility of extracting global consensus. Although the paper lacks further theoretical validation, the authors emphasize that due to the inherent difficulty in interpretability within model merging, their claims can still be supported experimentally.

**Essential References Not Discussed:**

I think the references in this paper are comprehensive.

**Experimental Designs Or Analyses:**

The main experiments consist of four parts: Visual multi-task, NLP multi-task, CB-EMS analysis, and Efficient-Aware Framework analysis. The core experiments largely validate CALM’s feasibility, showing strong performance on both visual and NLP tasks. The analysis sufficiently examines the robustness and effectiveness of each proposed module. One question: since the CB-EMS method does not exhibit stable sampling rates, how should sampling be handled in practice?

**Methods And Evaluation Criteria:**

The three contributions of this method hold good novelty and practical value for the model merging community. CB-EMS can serve as a general and feasible approach for unsupervised data in model merging; the Efficient-Aware Framework introduces a new serialized merging approach; while binary mask training follows Localize-and-Stitch, employing global task optimization is indeed more reasonable. Therefore, this method demonstrates clear advantages over previous approaches. If the authors further develop interpretable theories, it would be more beneficial to the community. For experiments, the paper provides sufficiently comprehensive tests, and its evaluation criteria align with classical model merging works.

**Other Comments Or Suggestions:**

Ensure minimal overlap among elements in figures, as exemplified in Figure 1.

**Other Strengths And Weaknesses:**

Strengths:
(1) The authors present a novel perspective by addressing the fundamental challenges of model merging, proposing a method that balances global task consensus with local parameter alignment.
(2) The framework of the proposed method is exceptionally clear, with each component being structurally complete and innovating upon existing model merging approaches in all three modules.
(3) The experimental results demonstrate substantial performance improvements over baselines, even compared to data-based methods.


Weaknesses:
(1) The sampling rates of CB-EMS in experiments appear inconsistent; how would this be resolved in practical applications?
(2) A detailed ablation study on global consensus is limited. The authors should explore the concept of global task consensus more deeply.

**Questions For Authors:**

See weakness

**Relation To Broader Scientific Literature:**

As for research on model merging, there are mainly two approaches. One is data-free methods, which do not require additional task data and rely on experience and strategies to merge models. The other is data-based methods, which utilize extra task information to achieve better merging results. Obviously, the CALM method falls into the latter category, but it fully learns from the former approaches by using task information to develop merging strategies. This combination offers a useful perspective for model merging research.

**Theoretical Claims:**

Eq.2-Eq.5 in the paper provide the derivation of CB-EMS, and Eq.6-Eq.9 outline the optimization process of Consensus-Aware Mask Optimization. There is no proof process, so no proof errors exist.

---

> ### Author Rebuttal · Authors · 2025-03-29
>
> ***Question 1: The sampling rates of CB-EMS in experiments appear inconsistent; how would this be resolved in practical applications?***
>
> ***Answer:***  Thanks for the practical comments. We would like to explain it as follows:
>
> - **CB-EMS Sampling Rates Show Cross-Task Generalizability.**
> Experimental results demonstrate that CALM exhibits consistent performance trends across language and visual benchmarks: accuracy first increases then decreases with higher sampling rates, peaking at 0.8 (language) and 0.9 (visual). This validates CALM's cross-domain adaptability and task-agnostic performance patterns.
>
> - **Optimal Sampling Rates Depend on Unsupervised Data Properties.**
> The choice of optimal sampling rates is dynamically determined by the scale and quality of unsupervised data. For small datasets, higher rates (0.8-0.9) help mitigate data scarcity through increased sample diversity. When individual task models show high prediction confidence on the unsupervised data, high sampling rates remain viable despite potential noise. Conversely, lower rates (0.4-0.5) are recommended for unreliable predictions to prioritize data credibility.
>
> - **Supervised Data Guides Sampling Rate Optimization.**
> Supervised data enables adaptive sampling: set sampling rate equal to task models' accuracy on the supervised set. This balances data trustworthiness and utilization by filtering noise while keeping useful data. **Below we demonstrate the performance improvements enabled by this method. Setting the sampling rate to match the accuracy of the supervised dataset significantly enhances data credibility.**
> ||SUN397|Cars|RESISC45|EuroSAT|SVHN|GTSRB|MNIST|DTD|
> |-|-|-|-|-|-|-|-|-|
> |Individual(Full Data)|75.3|77.7|96.1|99.7|97.5|98.7|99.7|79.4|
> |Sampling Rate|75.3|77.7|96.1|99.7|97.5|98.7|99.7|79.4|
> |Individual(Sampled Data)|88.3|87.3|97.4|99.9|98.5|99.2|99.8|87.5|
>
> ---
>
> ***Question 2:  A detailed ablation study on the individual components of CALM.***
>
> ***Answer:*** Thanks for the very constructive comments.
>
> - **Ablation Study of CB-EMS.**
> **As shown in Figure 5, we present ablation study results of CB-EMS.** The experimental results demonstrate that adding the CB-EMS method achieves performance comparable to supervised learning results and is significantly better than unsupervised EM methods using entropy.
> **Below, we provide the "Class-balanced" ablation results.** Incorporating the Class-balanced component effectively avoids class imbalance and shows significant performance improvement.
> |Method|SUN397|Cars|RESISC45|EuroSAT|SVHN|GTSRB|MNIST|DTD|**Avg Acc**|
> |-|-|-|-|-|-|-|-|-|-|
> |CB-EMS|72.6|74.8|91.9|98.6|95.2|96.4|99.1|72.8|**87.7**|
> |EMS|70.6|73.0|88.2|93.3|88.1|75.9|93.7|70.0|**81.5**|
>
> - **Ablation Study of Efficient-Aware Framework.**
> **We conduct additional ablation studies by adding the Efficient-Aware Framework (EAF) to existing model merging methods and removing it from CALM.** EAF has no effect on order-insensitive methods (e.g., Task Arithmetic) but effects order-sensitive methods (AdaMerging). Without EAF, CALM reduces to randomly selecting a task for mask learning, with results as follows.
> **Existing methods exhibit significant performance degradation under the Efficient-Aware Framework** since they require simultaneous consideration of all task vectors.
> **Without this framework, CALM still maintains robust performance.** Notably, CALM still requires EAF for large-scale task merging.
> |Method|SUN397|Cars|RESISC45|EuroSAT|SVHN|GTSRB|MNIST|DTD|**Avg Acc**|
> |-|-|-|-|-|-|-|-|-|-|
> |AdaMerging (+EAF)|70.7|55.3|61.3|66.5|74.0|58.1|99.1|36.2|**65.2**|
> |AdaMerging (-EAF)|64.5|68.1|79.2|93.8|87.0|91.9|97.5|59.1|**80.1**|
> |CALM (+EAF)|72.6|74.8|91.9|98.6|95.2|96.4|99.1|72.8|**87.7**|
> |CALM (-EAF)|71.9|74.1|91.6|98.7|95.2|96.2|98.9|71.2|**87.3**|
>
> - **Ablation Study of Consensus-Aware Mask Optimization.**
> Consensus-Aware Mask Optimization contributes most performance gains as our core component and cannot be fully ablated. We instead conduct ablation studies on our core insights and key components.
>     - **Ablation Study of Localized Information and Global Consenus**
> As core insights, CALM degenerates into Localize-and-Stitch (extracting localized task-specific information) when focusing solely on localized patterns, and degenerates into TW AdaMerging (applying identical masks to all parameters) when only global consensus is considered.
>     - **Ablation Study of Regularization**
> Evaluating the impact of regularization in the optimization equation.
>     - **Ablation Study of Binary Mask**
> Please refer to **Reviewer Rsjn Question 2**.
> |Method|SUN397|Cars|RESISC45|EuroSAT|SVHN|GTSRB|MNIST|DTD|**Avg Acc**|
> |-|-|-|-|-|-|-|-|-|-|
> |Local-only|67.2|68.3|81.8|89.4|87.9|86.6|94.8|62.9|**79.9**|
> |Global-only|58.0|53.2|68.8|85.7|81.1|84.4|92.4|44.8|**71.1**|
> |W/o Regularization|70.1|72.4|90.8|97.6|94.2|95.3|98.2|70.5|**86.3**|
> |CALM|72.6|74.8|91.9|98.6|95.2|96.4|99.1|72.8|**87.7**|

---

> > ### Comment · Reviewer_oEok · 2025-04-03
> >
> > Thank you for your detailed answer. I agree with the concept of integrating local information with global consensus in model merging, which offers valuable insights for the field. The authors provide comprehensive ablation studies that validate the effectiveness of their approach and address my concerns. I have no further questions and recommend maintaining my original score.

---

### Official Review · Reviewer_Rsjn · 2025-03-10

**Overall Recommendation:** 3

**Summary:**

This paper focuses on model merging in multi-task learning, aiming to identify locally shared information with global task consensus while addressing existing limitations of parameter conflict in global information and diluted local features during merging.
The method comprises three components: 1) Class-Balanced Entropy Minimization Sampling constructs a reliable unsupervised dataset; 2) An Efficiency-Aware Framework enables resource-effective model merging; 3) Consensus-Aware Mask Optimization designs mask refinement with global consensus.
Experiments on both vision and language datasets demonstrate the method's effectiveness, outperforming state-of-the-art approaches.

**Claims And Evidence:**

Most claims made in the submission are clear. However, the authors should further clarify the exact definition of "global consensus" and explain how CALM specifically achieves it.

**Essential References Not Discussed:**

The author should add a comparison between CALM and some of the latest model merging-related papers, like [1] and [2].

[1] Mitigating Parameter Interference in Model Merging via Sharpness-Aware Fine-Tuning
[2] Model merging with SVD to tie the Knots

**Experimental Designs Or Analyses:**

The experiments include 8 visual classification tasks and 12 NLP tasks, which aligns with most model merging methods.  However, the lack of analysis on the binary mask—such as whether it captures global consensus or the properties of this consensus—weakens the support for the authors’ key insight.

**Methods And Evaluation Criteria:**

Yes

**Other Comments Or Suggestions:**

Please refer to Other Strengths And Weaknesses and Questions For Authors

**Other Strengths And Weaknesses:**

Other Strengths:
1. The paper is well-structured and well written, and the figures and tables are detailed, clearly conveying the authors' insights and methodology.
2. I agree that the issue the authors focus on is significant in model merging.
3. The proposed method achieved state-of-the-art experimental results, and the experiments are thorough.

Other Weaknesses:
1. The experiments based on language multi-task benchmarks could be more comprehensive to check if the findings align with those in visual benchmarks.

**Questions For Authors:**

The following are the questions and concerns I hope the authors can address further:
1. The authors need to provide a clearer definition and explanation of global task consensus to establish the core idea of this work.
2. Can CALM adapt to existing frameworks without an efficiency-aware framework?
3. Does the binary mask capture global consensus? The authors need to explore this further.

**Relation To Broader Scientific Literature:**

The main contribution of this paper lies in applying the global optimization approach from Adamerging to localized model merging methods, such as Ties-Merging, and Localize-and-Stitch, thereby avoiding the issue where these data-free methods struggle to precisely capture task-specific characteristics.

**Theoretical Claims:**

No errors in theoretical claims. However, whether character subscripts should be italicized needs consistency.

---

> ### Author Rebuttal · Authors · 2025-03-28
>
> ***Question 1: The authors need to provide a clearer definition and explanation of global task consensus to establish the core idea of this work.***
>
> ***Answer:***  Thanks for your valuable question. We clarify the concept of global task consensus as follows:
>
> - **Definition:** Global task consensus refers to shared and generalizable knowledge patterns or parameter adaptation directions among multiple independently fine-tuned models during model merging.
> - **Key Properties:**
>   - **Cross-Task Validity:** Consistent parameter adaptation directions across tasks during fine-tuning.
>   - **Task-Specific Feature Preservation:** Avoidance of significant performance degradation on any task after model merging.
> - **Validation:** Due to length limitations, we provide theoretical error analysis to validate its effectiveness. For details, please refer to **Reviewer y6gA Question 1**.
>
> ---
>
> ***Question 2: Can CALM adapt to existing frameworks without an efficiency-aware framework?***
>
> ***Answer:***  Thank you for your valuable question.
>
> The efficiency-aware framework enables CALM to achieve efficient model merging with strong scalability. Our method can also adapt to scenarios without this framework, maintaining compatibility with existing approaches. Below we present two feasible solutions:
>
> - **Single-Task Merging: Randomly select one task for model merging.** As shown in Figure 6, the results demonstrate that applying CALM to a single task achieves comparable or even better performance than sequential multi-task merging. Thus, CALM can be applied to a single task without requiring sequential merging.
> - **Multi-Task Simultaneous Merging: Utilize multiple binary masks to merge all tasks simultaneously.**  We can slightly modify the CALM method to align with existing model merging frameworks, with the following optimization objective. Within a single optimization equation, we optimize a binary mask for each task and enforce that the sum of binary masks equals an all-ones matrix. This allows simultaneous extraction of information from all tasks while maintaining global consensus. During optimization, we still employ real-valued masks $R$ and classify each parameter to tasks via softmax.
> $$
> \min_{\{M\}} \ \sum_{t=1}^T\mathcal{L}(\theta_{mtl})+\alpha\sum_{t=1}^T||M_t||_1
> $$
>
> $$
> s.t. \ \theta_{mtl}=\theta_{pre}+\sum_{t=1}^TM_t\odot\tau_t,\  \sum_{t=1}^TM_t=\textbf{1}_{n \times n}.
> $$
>
> ---
>
> ***Question 3: Does the binary mask capture global consensus?***
>
> ***Answer:***  Thanks for the practical comment.
>
> In Appendix D, we provide visualizations and analysis of the binary mask properties. Here, we present a concise experiment to demonstrate the effectiveness of binary mask.
>
> For each task, we apply CALM with the following definitions: $\theta_{pre}$ denotes the pre-trained model, $\tau_{1}$ is the task vector from the remaining seven tasks via task arithmetic, $\tau_{2}$ is the target task vector, and $M$ is the optimized binary mask. We evaluate the target task performance under five configurations:
>
> |Model Parameters|SUN397|Cars|RESISC45|EuroSAT|SVHN|GTSRB|MNIST|DTD|
> |-|-|-|-|-|-|-|-|-|
> |$\theta_{pre}$|62.3|59.6|60.3|45.7|31.6|32.6|48.3|44.4|
> |$\theta_{pre}+\tau_1$|48.8|54.9|46.3|35.4|46.7|32.9|74.8|35.0|
> |$\theta_{pre}+(1-M)\odot\tau_1$|60.4|64.8|69.8|85.6|76.5|68.6|89.5|54.0|
> |$\theta_{pre}+\tau_2$|79.2|77.7|96.1|99.8|97.5|98.7|99.7|79.4|
> |$\theta_{pre}+M\odot\tau_1$|69.9|69.5|80.0|88.1|72.7|71.7|90.4|59.7|
> |$\theta_{pre}+(1-M)\odot\tau_1+M\odot\tau_2$|71.9|73.9|88.3|96.9|93.4|93.9|98.0|69.0|
>
> CALM extracts task vectors with approximately 5% parameters. We have the following conclusions:
> - **Interference suppression** (Rows 2-3): Removing 5% parameters from $\tau_{1}$ via $(1-M)\odot\tau_{1}$ significantly improves target task adaptation, demonstrating effective identification of interference parameters.
> - **Task-specific preservation** (Rows 4-5): Confining $\tau_{2}$ to 5% masked parameters $M\odot\tau_{2}$ maintains strong performance, indicating precise capture of essential transfer signals.
> - **Conflict-free merging** (Rows 3,5,6): Simultaneous application of $(1-M)\odot\tau_{1}$ and $M\odot\tau_{2}$ achieves performance gains without parameter conflicts, verifying CALM's effective multi-task merging capability.
>
> ---
>
> ***Question 4: Additional experiments of language multi-task benchmarks.***
>
> ***Answer:*** Thanks for the very constructive comments.
>
> **CALM demonstrates consistent behavior across language and visual benchmarks.** We give an example, shown in the table below. Both domains exhibit nearly identical trends in average accuracy versus CB-EMS sampling rates - initially increasing then decreasing with higher sampling. Additional experiments confirm this cross-domain consistency.
>
> |Sampling Rates|0.1|0.2|0.3|0.4|0.5|0.6|0.7|0.8|0.9|1.0|
> |-|-|-|-|-|-|-|-|-|-|-|
> |language benchmarks|80.1|80.8|81.0|81.2|81.2|81.3|81.3|**81.7**|81.4|81.5|
> |visual benchmarks|83.8|84.9|85.7|86.1|86.4|86.8|87.1|87.4|**87.7**|87.5|

---

> > ### Comment · Reviewer_Rsjn · 2025-04-05
> >
> > Thanks for the rebuttal. Most of my concerns have been addressed. I will keep my score

---

### Official Review · Reviewer_y6gA · 2025-03-11

**Overall Recommendation:** 3

**Summary:**

Localized Information with Global Consensus: CALM proposes a method to extract localized parameters that align with global task consensus, ensuring that the merged model maintains effectiveness across all tasks.

* A new sampling technique that leverages unsupervised data more effectively by balancing class representation and minimizing entropy.

* A scalable and efficient framework for merging models sequentially, reducing computational complexity while maintaining performance.

* A method to optimize binary masks that align with global task consensus, enabling conflict-free merging of task-specific parameters.

* The authors demonstrate the superiority of CALM through extensive experiments on both vision and NLP tasks, showing that it outperforms existing global-aware and localized-aware methods, and approaches the performance of traditional MTL without the need for retraining.

**Claims And Evidence:**

Yes, most claims are support by empirical evidence.

**Essential References Not Discussed:**

N/A

**Experimental Designs Or Analyses:**

Yes, the experimental designs are sound.

**Methods And Evaluation Criteria:**

The evaluation criteria follows existing works and is reasonable in practice.

**Other Comments Or Suggestions:**

N/A

**Other Strengths And Weaknesses:**

**Strengths**:

* The authors provide extensive experimental results across multiple datasets, demonstrating that CALM consistently outperforms existing baselines and achieves performance close to traditional MTL.

* The efficient-aware framework is a practical contribution, especially in scenarios where tasks are completed asynchronously or computational resources are limited. The sequential merging approach reduces the complexity of merging multiple tasks.




**Weaknesses**:

* While the empirical results are strong, the paper lacks a deeper theoretical analysis of why localized information with global consensus leads to better performance. A more formal theoretical framework or proof could strengthen the paper.

* Although the paper includes some ablation studies (e.g., comparing CB-EMS with other sampling methods), it would benefit from a more detailed analysis of the individual components of CALM (e.g., the impact of the efficient-aware framework vs. the consensus-aware mask optimization).


* The paper does not discuss the sensitivity of CALM to hyperparameters (e.g., the regularization parameter
$\lambda$ or the sampling rate for CB-EMS). A discussion on how sensitive the method is to these choices would be useful for practitioners.


* efficient-aware->   efficiency-aware?

**Questions For Authors:**

N/A

**Relation To Broader Scientific Literature:**

N/A

**Theoretical Claims:**

No theoretical claims.

---

> ### Author Rebuttal · Authors · 2025-03-27
>
> ***Question 1: A more formal theoretical analysis of the effectiveness of localized information with global consensus***
>
> ***Answer 1***: Thanks for the inspiring question.  We perform a theoretical analysis of the following three aspects based on error.
>
> - **Localized information reduces interference in model merging. Such interference arises from intra-task noise and conflicts across task vectors (e.g., cancellation due to opposing parameter values).**
> Denote the interference by $\epsilon$, so that task vector$$\tau_j = \Delta\tau_j+\epsilon,$$ where $\Delta\tau_j$ is the effective information for the global optimal update (nonzero in dimensions $I$) and $\epsilon$ is concentrated in the complementary set $J$. Define the binary mask $M_L$ such that $(M_L)_i=1$ if $i \in I$ and
> $(M_L)_i=0$ if $i \in J$. Thus, the merged task vector becomes $$\Delta \tau_L = M_L\odot\tau_j = \Delta\tau_j+M_L\odot\epsilon,$$
> reducing the interference energy from $||\epsilon||^2$ to $E_L=||M_L\odot\epsilon||^2.$
> Although methods like Ties-Merging and Consensus TA aim to lower interference, their effectiveness is limited by the absence of global task information.
>
> - **Global consensus enables the update to closely align with the optimal direction common to all tasks, reducing the overall loss.**
> For all visible tasks $S_v$, we utilize global consensus to optimize a mask $M_G$ by minimizing
> $$
> \min_{M_G} \sum_{t_v \in S_v} L_{t_v}(\theta_{pre} + \tau_{seq}^{(j-1)} + M_G \odot \tau_j).
> $$
> Assuming a quadratic approximation near $\theta_{pre} + \tau_{seq}^{(j-1)}$, we have
> $$
> L_{t_v}(\theta_{pre} + \tau_{seq}^{(j-1)} + M_G \odot \tau_j) \approx L_{t_v}(\theta_{pre} + \tau_{seq}^{(j-1)}) + \langle g_{t_v}, M_G \odot \tau_j \rangle + \frac{1}{2}(M_G \odot \tau_j)^T H_{t_v} (M_G \odot \tau_j),
> $$
> where $H_{t_v}$ is the Hessian. Assume $H_{t_v}$ is the identity matrix and $\Delta\tau_j$ denote the global optimal update. The update error is
> $$
> E_{G} = ||M_G \odot \tau_j - \Delta\tau_j||^2.
> $$
> The gain from global consensus is defined as
> $$
> \Delta_{G} = ||\tau_j - \Delta\tau_j||^2 - ||M_G \odot \tau_j - \Delta\tau_j||^2.
> $$
> The optimization process shows that global consensus effectively reduces the overall loss.
>
> - **Our CALM method integrates localized information and global consensus to effectively mitigate local parameter interference while capturing information beneficial for global tasks.**
> Let the obtained binary mask be $M^*$; then the update error is
> $$
> E_{CALM} = ||M^* \odot \tau_j - \Delta \tau_j||^2.
> $$
> Based on the optimization objective, the error can also be expressed as
> $$
> E_{CALM} = \min(E_{loc}, E_{glob}) - \Delta_{syn},
> $$
> where $\Delta_{syn} > 0$ represents the additional error reduction from joint optimization.
>
> **Thus, CALM achieves both local and global gains, effectively reducing the overall error.**
>
> ---
>
> ***Question 2: Additional ablation studies on the individual components of CALM.***
>
> ***Answer:*** Thank you for your valuable question.
>
> Due to length limitations, we will present additional ablation study results in **Reviewer oEok Question 2**.
>
> ---
>
> ***Question 3: The sensitivity of CALM to hyperparameters (the regularization parameter $\lambda$ and the sampling rate for CB-EMS).***
>
> ***Anwser:*** Many thanks for the insightful suggestions.
>
> Below are the model merging results on eight vision datasets under different regularization parameters $\lambda$ and sampling rates.
>
> |$\lambda$|SUN397|Cars|RESISC45|EuroSAT|SVHN|GTSRB|MNIST|DTD|Avg Acc|
> |-|-|-|-|-|-|-|-|-|-|
> |0.5|72.9|75.2|92.3|99.0|95.3|96.3|99.1|72.9|**87.8**|
> |2|72.5|74.8|91.8|98.7|95.3|96.3|99.1|72.1|**87.6**|
> |1|72.6|74.8|91.9|98.6|95.2|96.4|99.1|72.8|**87.7**|
>
> CALM is insensitive to $\lambda$, with merging performance remaining stable.
>
> **Thus, $\lambda$ between 0.5 and 2 is acceptable**.
>
> |Sampling Rate|SUN397|Cars|RESISC45|EuroSAT|SVHN|GTSRB|MNIST|DTD|Avg Acc|
> |-|-|-|-|-|-|-|-|-|-|
> |0.1|69.7|69.0|88.6|97.1|93.9|94.1|96.8|61.4|**83.8**|
> |0.2|70.8|70.3|89.2|97.4|94.0|94.7|97.2|65.4|**84.9**|
> |0.3|71.5|71.8|89.9|97.9|94.3|95.0|97.5|67.5|**85.7**|
> |0.4|71.9|72.4|90.2|98.0|94.3|95.1|97.8|69.1|**86.1**|
> |0.5|72.2|73.3|90.8|98.2|94.6|95.6|98.0|68.8|**86.4**|
> |0.6|72.6|73.5|91.0|98.2|94.5|95.6|98.1|70.9|**86.8**|
> |0.7|72.4|74.5|91.2|98.4|94.8|95.8|98.2|71.7|**87.1**|
> |0.8|72.7|74.9|91.7|98.6|95.0|96.0|98.5|71.8|**87.4**|
> |0.9|72.6|74.8|91.9|98.6|95.2|96.4|99.1|72.8|**87.7**|
> |1.0|72.9|75.3|91.8|98.6|95.2|95.9|98.8|71.8|**87.5**|
>
> Performance first increases then decreases with the sampling rate, peaking at 0.9. This aligns with our theory.
>
> **In practice, the optimal sampling rate depends on data quantity and quality; for high-quality data, a higher rate (e.g., 0.8–0.9) is preferred, otherwise a lower rate is recommended**.
>
> ---
>
> ***Question 4: efficient-aware-> efficiency-aware.***
>
> ***Answer:*** Thank you for your constructive advice.
>
> **We will seriously consider making changes in the final version.**

---

### Decision · Program_Chairs · 2025-05-01

**Decision:**

Accept (poster)

**Comment:**

This paper introduces CALM, a novel model merging approach for multi-task learning that identifies localized parameters aligned with global task consensus. Extensive experiments on vision and NLP benchmarks demonstrate CALM’s superiority over existing methods, approaching the performance of traditional multi-task learning without retraining.

CALM makes significant empirical and practical contributions to model merging, with novel components (CB-EMS, consensus masking) advancing the field, strong scalability demonstrated across 26+ tasks, and thorough rebuttals addressing key concerns.

While deeper theoretical analysis would strengthen the work, all reviewers think that the consistent experimental gains and methodological innovations justify acceptance.